# eIF2B activator prevents neurological defects caused by a chronic integrated stress response

Yao Liang Wong[1†], Lauren LeBon[1], Ana M Basso[2], Kathy L Kohlhaas[2], Arthur L Nikkel[2], Holly M Robb[2], Diana L Donnelly-Roberts[2], Janani Prakash[2], Andrew M Swensen[2], Nimrod D Rubinstein[1], Swathi Krishnan[1], Fiona E McAllister[1], Nicole V Haste[1], Jonathon J O'Brien[1], Margaret Roy[1], Andrea Ireland[1], Jennifer M Frost[2], Lei Shi[2], Stephan Riedmaier[2], Kathleen Martin[1], Michael J Dart[2], Carmela Sidrauski[1]*

[1]Calico Life Sciences LLC, South San Francisco, United States; [2]AbbVie, North Chicago, United States

**Abstract** The integrated stress response (ISR) attenuates the rate of protein synthesis while inducing expression of stress proteins in cells. Various insults activate kinases that phosphorylate the GTPase eIF2 leading to inhibition of its exchange factor eIF2B. Vanishing White Matter (VWM) is a neurological disease caused by eIF2B mutations that, like phosphorylated eIF2, reduce its activity. We show that introduction of a human VWM mutation into mice leads to persistent ISR induction in the central nervous system. ISR activation precedes myelin loss and development of motor deficits. Remarkably, long-term treatment with a small molecule eIF2B activator, 2BAct, prevents all measures of pathology and normalizes the transcriptome and proteome of VWM mice. 2BAct stimulates the remaining activity of mutant eIF2B complex in vivo, abrogating the maladaptive stress response. Thus, 2BAct-like molecules may provide a promising therapeutic approach for VWM and provide relief from chronic ISR induction in a variety of disease contexts.
DOI: https://doi.org/10.7554/eLife.42940.001

*For correspondence:
carmelas@me.com

†These authors contributed equally to this work

## Introduction

eIF2B is the guanine nucleotide exchange factor (GEF) for the GTPase and initiation factor eIF2 and modulation of its activity is central to regulation of protein synthesis rates in all eukaryotic cells. GTP-bound eIF2 associates with the initiator methionyl tRNA and this ternary complex (eIF2-GTP-Met-tRNAi) delivers the first amino acid to the ribosome. GTP is hydrolyzed and eIF2-GDP is released, requiring reactivation by eIF2B to enable a new round of protein synthesis (*Hinnebusch and Lorsch, 2012*). Four stress-responsive kinases (PERK, HRI, GCN2 and PKR) that detect diverse insults converge on phosphorylation of eIF2 at serine 51 of the α subunit (eIF2α). Phosphorylation converts eIF2 from a substrate of eIF2B into its competitive inhibitor, triggering the integrated stress response (ISR), which reduces translation initiation events and decreases global protein synthesis (*Krishnamoorthy et al., 2001*; *Yang and Hinnebusch, 1996*). Concomitantly, the ISR induces the translation of a small set of mRNAs with special sequences in their 5' untranslated regions, including the transcription factor ATF4 (*Harding et al., 2000*; *Watatani et al., 2008*). ATF4 triggers a stress-induced transcriptional program that is thought to promote cell survival during mild or acute conditions but can contribute to pathological changes under severe or chronic insults (*Pakos-Zebrucka et al., 2016*).

Vanishing White Matter (VWM; OMIM 603896) is a rare, autosomal recessive leukodystrophy that is driven by mutations in eIF2B (*Leegwater et al., 2001*; *van der Knaap et al., 2002*). The disease is

**eLife digest** Cells must be able to respond to their changing environment in order to survive. When cells encounter particularly unfavorable conditions, they often react by activating a so-called 'stress' response. A group of proteins collectively known as eIF2B helps to regulate this response.

In a severe neurological condition called Vanishing White Matter (VWM), the genes that produce the eIF2B proteins contain mutations that make eIF2B less active. As a result, certain cells in people with VWM are always stressed.

Six years ago, researchers discovered a molecule that boosts the activity of eIF2B. In 2018, they found that it also works on various mutant forms of eIF2B found in VWM. The molecule had so far only been tested in biochemical laboratory experiments. Now, Wong et al. – including some of the researchers involved in the 2018 study – have tested whether an improved version of the molecule treats VWM in mice.

The trial treatment successfully halted all signs of the disease in the mice. The molecule blunted the persistent stress response of the cells in the brain and spinal cord, primarily in a cell type that is severely affected by the human form of VWM. Cells in other parts of the body were spared.

Overall, the results of the experiments suggest that an eIF2B activator may prove to be an effective treatment for VWM in humans. It could similarly be used to treat other conditions that activate this abnormal cell stress response. The molecule Wong et al. used is not suitable for use in humans, so work is continuing to find a suitable variant.

DOI: https://doi.org/10.7554/eLife.42940.002

characterized by myelin loss, progressive neurological symptoms such as ataxia, spasticity, cognitive deterioration and ultimately, death (*Schiffmann et al., 1994*; *van der Knaap et al., 1997*). Age of VWM onset is variable and predictive of disease progression, ranging from severe prenatal/infantile onset leading to death in months (as in the case of 'Cree leukoencephalopathy') to slower-progressing adult onset presentations (*Fogli et al., 2002*; *Hamilton et al., 2018*; *van der Knaap et al., 2006*). Because eIF2B is essential, VWM mutations are restricted to partial loss-of-function in any of the five subunits of the decameric complex (*Fogli et al., 2004*; *Horzinski et al., 2009*; *Li et al., 2004*; *Liu et al., 2011*). Nearly 200 different mutations have been catalogued to date in the Human Gene Mutation Database, which occur as homozygotes or compound heterozygotes with a different mutation in each allele of the same subunit. Reduction of eIF2B activity is analogous to its inhibition by phosphorylated eIF2$\alpha$, thus it is congruent and compelling that ISR activation has been observed in VWM patient post-mortem samples (*van der Voorn et al., 2005*; *van Kollenburg et al., 2006*).

We previously showed that a range of VWM mutations destabilize the eIF2B decamer, leading to compromised GEF activity in both recombinant complexes and endogenous protein from cell lysates (*Wong et al., 2018*). We demonstrated that the small molecule eIF2B activator ISRIB (for ISR inhibitor) stabilized the decameric form of both wild-type (WT) and VWM mutant complexes, boosting their intrinsic activity. Notably, ISRIB bridges the symmetric dimer interface of the eIF2B central core, acting as a molecular stapler (*Tsai et al., 2018*; *Zyryanova et al., 2018*). In addition, we showed that ISRIB attenuated ISR activation and restored protein synthesis in cells carrying VWM mutations (*Wong et al., 2018*).

Although we found that ISRIB rescued the stability and activity of VWM mutant eIF2B in vitro, the ability of this class of molecules to prevent pathology in vivo remained an unanswered question. Knock-in mouse models of human VWM mutations have been characterized, and the severe mutations recapitulate key disease phenotypes such as progressive loss of white matter with concomitant manifestation of motor deficits (*Dooves et al., 2016*; *Geva et al., 2010*). Here, we generate an improved tool molecule and demonstrate that sustained eIF2B activation blocks maladaptive induction of the ISR and prevents all evaluated disease phenotypes in a VWM mouse model.

## Results

### 2BAct is a novel eIF2B activator with improved in vivo properties in rodents

To interrogate efficacy in vivo, we sought a small molecule eIF2B activator with improved solubility and pharmacokinetics relative to ISRIB. To that end, we synthesized 2BAct, which has a differentiated bicyclo[1.1.1]pentyl core and, unlike ISRIB, is no longer symmetric (*Figure 1A*). 2BAct is a highly selective eIF2B activator and exhibited similar potency to ISRIB in a cell-based reporter assay that measures ISR attenuation (*Figure 1—figure supplement 1A* and *Supplementary file 1A*). 2BAct is able to penetrate the central nervous system (CNS) (unbound brain/plasma ratio ~0.3) and also demonstrated dose-dependent oral bioavailability using an aqueous-based vehicle (*Supplementary file 1B*). Additionally, 2BAct is well-suited for formulation in diet, enabling long-term dosing without effects on body weight in WT mice (*Figure 1—figure supplement 1B*). The molecule was well-tolerated in the animal studies described here, and did not elicit any relevant effects in a rat cardiovascular (CV) safety study; however, significant anomalies were observed in a dog CV model. This CV safety liability makes this particular molecule unsuitable for human dosing.

### 2BAct normalized body weight gain in VWM mice

We generated a previously described mouse model of VWM that harbors the severe 'Cree leukoencephalopathy' mutation, $Eif2b5^{R191H/R191H}$ (hereafter referred to as R191H; *Figure 1—figure supplement 2*) (*Dooves et al., 2016*). The homologous human $Eif2b5^{R195H}$ mutation causes an infantile-onset, rapidly progressing form of VWM (*Black et al., 1988*). R191H mice recapitulate many aspects of the human disease, such as spontaneous myelin loss, progressive ataxia, motor skill deficits, and shortened lifespan (*Dooves et al., 2016*). We selected this severe disease allele for in vivo studies as pharmacological efficacy in this model is a stringent test for eIF2B activators and, mechanistically, should generalize to milder mutations as seen in vitro (*Wong et al., 2018*). We confirmed that primary fibroblast lysates from R191H embryos had lower GEF activity than WT lysates, and that 2BAct enhanced this activity threefold ($EC_{50}$ = 7.3 nM; *Figure 1—figure supplement 1C–D*).

To test efficacy, we undertook a 21-week blinded treatment study with 2BAct and measured intermediate and terminal phenotypes associated with disease progression in R191H mice (*Figure 1B*). 2BAct was administered orally by providing mice with the compound incorporated in rodent meal. This dosing regimen provided unbound brain exposures 15-fold above the in vitro $EC_{50}$ at the end of the study, ensuring saturating coverage of the target.

At the initiation of the study (6–11 weeks old), WT and R191H males had similar body weights (*Figure 1C* and *Figure 1—figure supplement 3A*). However, 1 week later, a significant difference emerged and continued to grow as R191H mice failed to gain weight (WT gain = 0.5 g/week vs. R191H gain = 0.08 g/week; $p<10^{-4}$). Remarkably, the body weight of 2BAct-treated R191H males caught up to WT mice 2 weeks after beginning dosing (8–13 weeks old), at which point their rate of weight gain equalized (WT gain = 0.5 g/week vs. R191H gain = 0.55 g/week; p = 0.24; *Figure 1C*). Similar results were observed in female mice (*Figure 1—figure supplement 4A*). As lack of weight gain appeared to be the first overt phenotype, rapid normalization by 2BAct was a promising prognostic sign of efficacious target engagement.

### 2BAct prevented the appearance of motor deficits in VWM mice

Longitudinal characterization revealed that R191H mice developed progressive, age-dependent strength and motor coordination deficits. From 8 to 19 weeks of age, R191H animals were not significantly different from WT in their performance on an inverted grid test of neuromuscular function. At 23 weeks, R191H mice showed a trend towards shorter hang times and at 26 weeks, this decrease was highly significant in both sexes (*Figure 1—figure supplement 3B*). In a beam-crossing test of balance and motor coordination, R191H mice were not significantly different from WT littermates at 8–19 weeks of age (*Figure 1—figure supplement 3C–D*). However, at 23 weeks of age, beam-crossing time was significantly increased in mutant animals, and they exhibited more foot slips and falls from the beam while crossing. The deficit in both parameters was exacerbated at 26 weeks, and some R191H animals completely failed to cross the beam within the trial cutoff of 30 s (*Figure 1—*

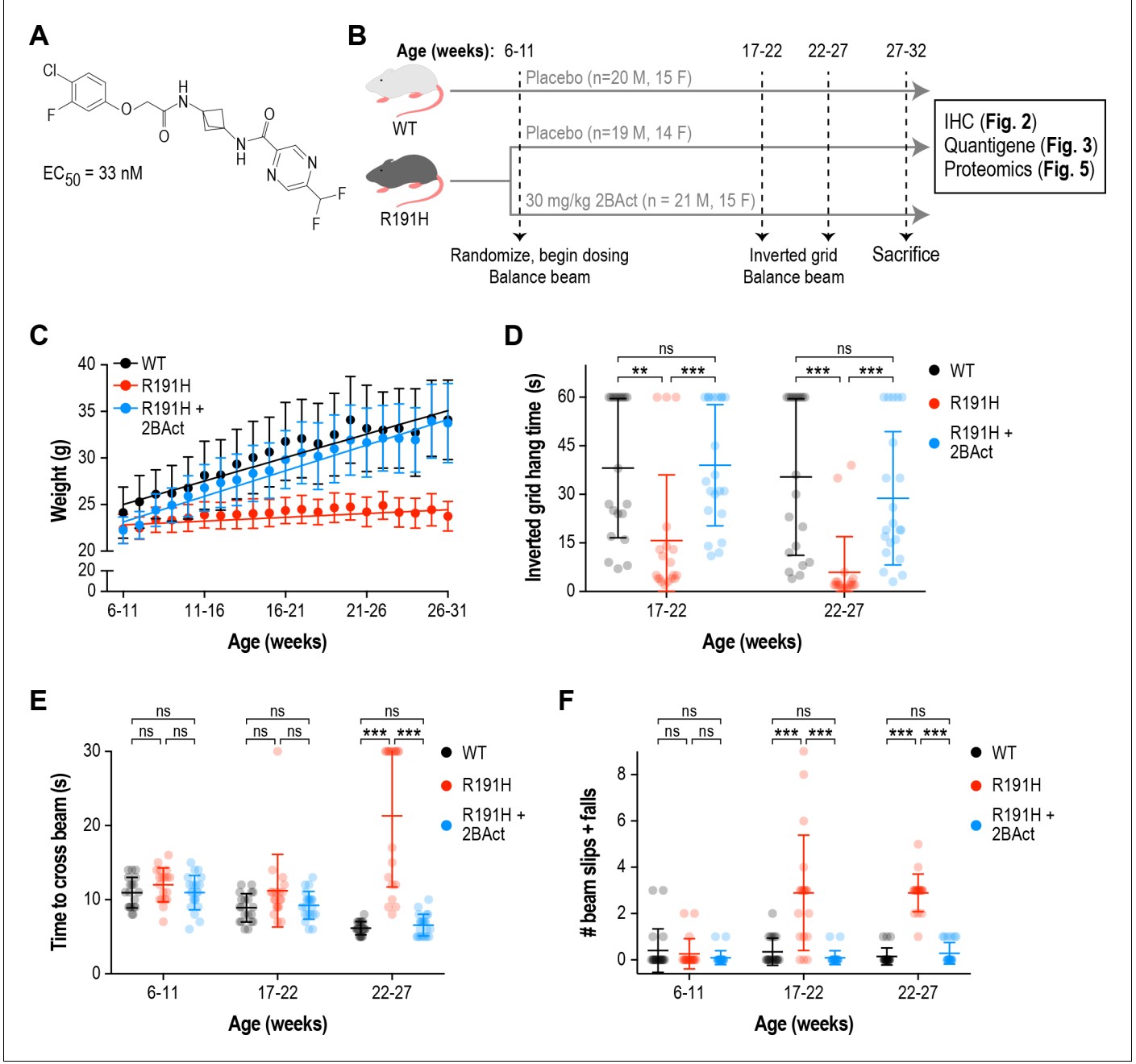

**Figure 1.** 2BAct normalized body weight gain and prevented motor deficits in male R191H mice. (A) Chemical structure of 2BAct and ATF4-luciferase reporter assay EC50. (B) Schematic of the 2BAct treatment experiment. Body weights were measured weekly for the duration of the experiment. (C) Body weight measurements of male mice along the study. Lines are linear regressions. At the 6–11 week time point when 2BAct treatment was initiated, body weights were not significantly different among the three conditions (p>0.05; two-way ANOVA with Holm-Sidak pairwise comparisons). R191H body weight was significantly lower at all subsequent time points. 2BAct-treated animals caught up to WT animals at the 8–13 week time point, and their weights were not significantly different thereafter. (D) Inverted grid test of muscle strength. Time spent hanging was measured up to a maximum of 60 s. (E–F) Beam-crossing assay to measure balance and motor coordination. Time to cross the beam was measured up to a maximum of 30 s (E), and the number of foot slips/falls was counted (F). For (C)-(F), N = 20 (WT), 19 (R191H) and 21 (R191H + 2 BAct) males were analyzed. Error bars are SD. For (D)-(F), *p<0.05; **p<0.01; ***p<10$^{-3}$; $^{ns}$p>0.05 by Mann-Whitney test with Bonferroni correction.

DOI: https://doi.org/10.7554/eLife.42940.003

The following figure supplements are available for figure 1:

**Figure supplement 1.** 2BAct is an eIF2B activator with similar potency to ISRIB.

DOI: https://doi.org/10.7554/eLife.42940.004

*Figure 1 continued on next page*

*Figure 1 continued*

**Figure supplement 2.** Generation of the R191H (*Eif2b5*[R191H (flox)/R191H (flox)]) mouse model.
DOI: https://doi.org/10.7554/eLife.42940.005
**Figure supplement 3.** R191H mice exhibited reduced body weight and age-dependent performance impairment in motor assays.
DOI: https://doi.org/10.7554/eLife.42940.006
**Figure supplement 4.** 2BAct normalized body weight gain and prevented motor deficits in female R191H mice.
DOI: https://doi.org/10.7554/eLife.42940.007

*figure supplement 3C*). These results are consistent with the original description of the R191H model (*Dooves et al., 2016*).

With baseline performance measured, we assessed the effect of 2BAct treatment on R191H motor skills. Placebo-treated R191H males had significantly reduced inverted grid hang times at both tested time points, as well as more coordination errors and time spent crossing the balance beam (*Figure 1D–F*). By contrast, 2BAct-treated R191H males were indistinguishable from WT in both assays. Full normalization was similarly observed in female animals (*Figure 1—figure supplement 4B–D*). Together, these data show that treatment of R191H animals with 2BAct prevented the progressive deterioration of motor function caused by VWM.

## 2BAct prevented myelin loss and reactive gliosis in VWM mice

VWM is a leukoencephalopathy defined by progressive loss of myelin. In patients with advanced disease, an almost complete loss of cerebral white matter is observed (*van der Knaap et al., 1997*). Similarly, in a previous characterization of the R191H mouse model, perturbed myelination and myelin vacuolization were noted in the brain beginning at 4–5 months of age (*Dooves et al., 2016*; *Klok et al., 2018*). Given the dramatic rescue of behavioral phenotypes by 2BAct, we performed immunohistochemical analysis to examine its effects on myelin and accompanying pathologies at the end of the treatment. We focused on two heavily myelinated regions, the corpus callosum and the spinal cord. Notably, severe spinal cord pathology has recently been reported in both VWM patients and R191H mice (*Leferink et al., 2018*).

As anticipated, R191H animals showed a clear reduction in myelin by Luxol Fast Blue staining of both regions (33% reduction in corpus callosum, $p<10^{-4}$; 58% reduction in cervical/thoracic spinal cord, $p<10^{-4}$; *Figure 2A–D*). Strikingly, 2BAct treatment maintained myelin levels at 91% and 85% of WT in the corpus callosum and spinal cord, respectively. Staining for myelin basic protein (MBP), an alternative measure of myelin content, corroborated these results in both regions (*Figure 2—figure supplement 1A–D*). As astrocytes have been implicated in the pathogenesis of VWM, we also stained for the astrocyte marker GFAP (*Dietrich et al., 2005*; *Dooves et al., 2016*). We found a significant increase in GFAP in both regions of placebo-treated R191H mice, which was fully normalized by 2BAct treatment (*Figure 2A–B* and *Figure 2—figure supplement 1C–D*).

Consistent with the literature, we noted a significant increase in Olig2 in R191H spinal cord (*Figure 2—figure supplement 1C–D*). Olig2 is a marker of the oligodendrocyte lineage, and its increase could indicate an attempt to compensate for the myelin loss (*Bugiani et al., 2011*). Additionally, we observed signs of reactive microglia in the placebo-treated R191H samples, as evidenced by a 15-fold increase in Iba-1 staining (*Figure 2C–D*). ATF3, an ISR target induced in the spinal cord during injury and inflammation, was also significantly increased (*Dominguez et al., 2010*; *Hossain-Ibrahim et al., 2006*). 2BAct treatment fully normalized all four of these markers (*Figure 2C–D* and *Figure 2—figure supplement 1C–D*). In an analysis of younger mice, we observed no significant differences in myelin, Iba-1 or ATF3 between R191H and WT spinal cord at the start of treatment (2 months of age; *Figure 2—figure supplement 2*). The time course of pathology was consistent with the emergence of motor deficits, and reflected the degenerative nature of VWM. Together, these results demonstrate that dosing with 2BAct before onset of histological signs prevents CNS pathology in a mouse model of VWM.

## A chronic ISR in the CNS of VWM mice is prevented by 2BAct

In all eukaryotic systems, ATF4 protein expression is regulated by the level of ternary complex in cells (*Harding et al., 2000*; *Mueller and Hinnebusch, 1986*; *Vattem and Wek, 2004*). We postulated that the decrease in eIF2B GEF activity brought about by the *Eif2b5*[R191H] mutation would

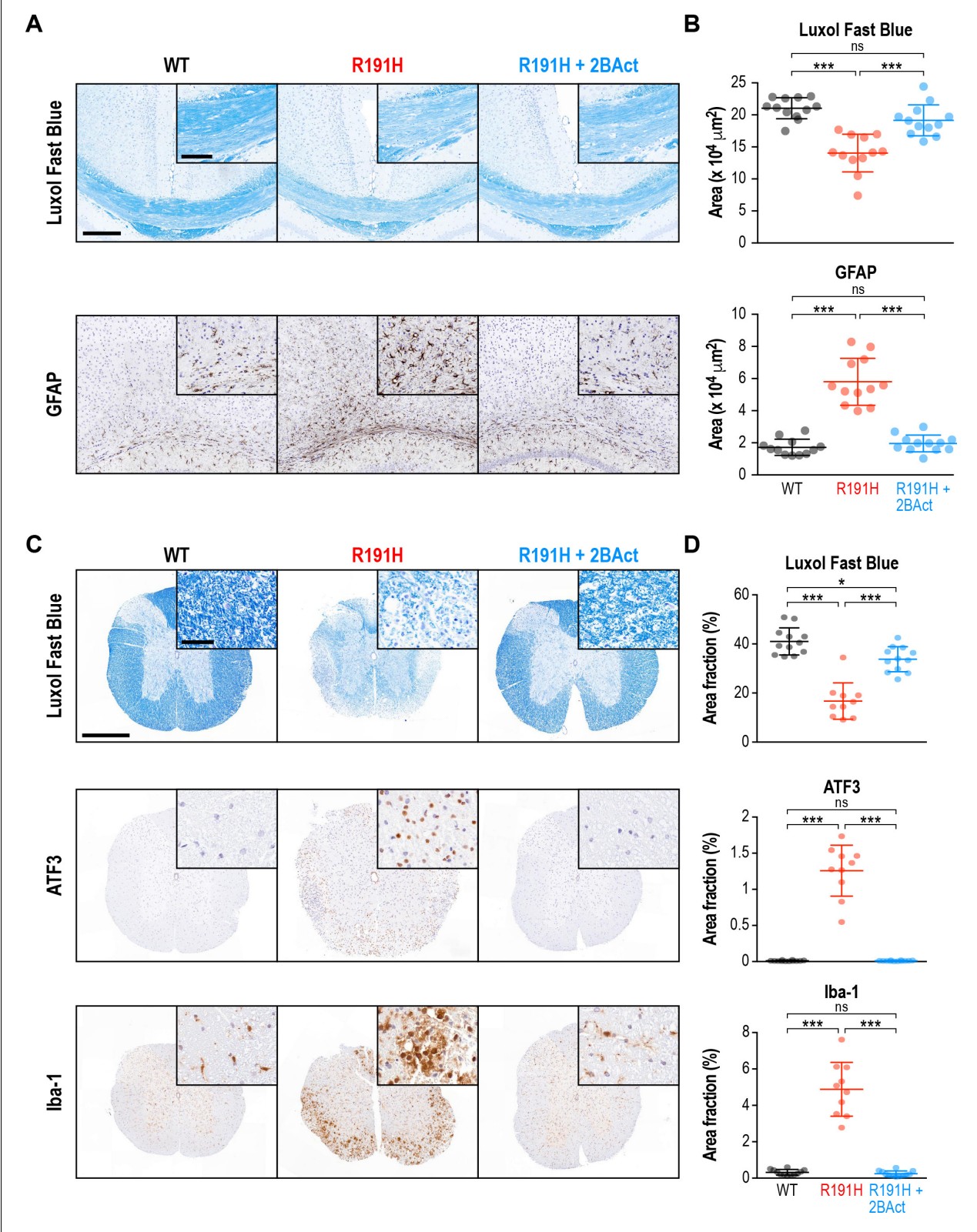

**Figure 2.** 2BAct prevented myelin loss and reactive gliosis in the brain and spinal cord of R191H mice. (**A**) Representative IHC images of the corpus callosum. Scale bar, 250 μm. Inset is magnified 2X. Inset scale bar, 100 μm. (**B**) Quantification of staining in (**A**). Area of positive staining expressed as μm². (**C**) Representative IHC images of the lower cervical/upper thoracic region of the spinal cord. Scale bar, 500 μm. Inset is magnified 6.8X. Inset scale

*Figure 2 continued on next page*

*Figure 2 continued*

bar, 50 µm. (D) Quantification of staining in (C). For (B) and (D), N = 12 mice/condition (6 males and six females combined; no significant sex differences were detected). Error bars are SD. *p<0.05; ***p<10$^{-4}$; $^{ns}$p>0.05 by 1-way ANOVA with Holm-Sidak pairwise comparisons.

DOI: https://doi.org/10.7554/eLife.42940.008

The following figure supplements are available for figure 2:

**Figure supplement 1.** 2BAct prevented myelin loss and reactive gliosis in the brain and spinal cord of R191H mice.

DOI: https://doi.org/10.7554/eLife.42940.009

**Figure supplement 2.** Age-dependent myelin loss and inflammation in the spinal cord of R191H mice.

DOI: https://doi.org/10.7554/eLife.42940.010

reduce levels of ternary complex, leading to upregulated ATF4 translation in R191H mice. In support of this, ISR activation has been reported in patient VWM postmortem samples (*van der Voorn et al., 2005*; *van Kollenburg et al., 2006*).

To evaluate ISR activity, we measured the expression of 15 transcripts previously identified as ATF4 target genes at three different time points (2.5, 5, and 7 months) during the lifespan and development of pathology. The ISR was robustly and consistently induced at these time points in the cerebellum, forebrain, midbrain and hindbrain of R191H animals (*Figure 3A* and *Figure 3—figure supplement 1A*). Significant upregulation of all targets except *Gadd34*, *Slc1a5* and *Gadd45a* was evident in cerebellum at 2.5 months; at later timepoints, these three transcripts also became significantly induced. The ISR signature was similar across all brain regions with *Atf5*, *Eif4ebp1* and *Trib3* being the most upregulated transcripts in the panel. Interestingly, we did not observe ISR induction in R191H mice at postnatal day 14 (*Figure 3—figure supplement 1B*). Thus, the ISR is activated sometime between 2 and 8 weeks of age through an as-yet unknown mechanism. Activation of this stress response preceded the appearance of pathology (myelin loss, gliosis and motor deficits) in VWM mice.

In addition to changes in transcript levels, we confirmed translational induction of ATF4 as well as the increase in protein levels of the negative regulator of cap-mediated mRNA translation EIF4EBP1 by Western blot analysis of 7-month-old R191H cerebellum lysates (*Figure 3B*). As expected for an ISR induced by eIF2B dysfunction rather than external stressors (see schematic in *Figure 3C*), we did not detect an increase in eIF2α phosphorylation in R191H brains.

We observed greater ISR induction in the spinal cord compared to the cerebellum (*Figure 3D*, compare red and brown points), with the transcription factor *Atf3* showing an additional 10-fold increase. The greater extent of myelin loss in the spinal cord, as well as astrocyte and microglial activation, suggests exacerbation of the phenotype due to increased ISR activation. Notably, treatment with 2BAct abolished ISR induction in both regions (*Figure 3B,D*). The striking attenuating effect of this molecule on the ISR is consistent with full rescue of GEF activity in vivo.

To determine whether our results would extend to other VWM mutations, we examined a second mouse model of VWM bearing an *Eif2b5*$^{R132H/R132H}$ mutation, which corresponds to the disease-causing human *Eif2b5*$^{R136H}$ allele (*Geva et al., 2010*). This model has a normal lifespan and exhibits very mild phenotypes in comparison to R191H mice. Nevertheless, we detected significant upregulation of two ISR targets, *Atf5* and *Eif4ebp1*, that was blocked by 4 weeks of treatment with 2BAct (*Figure 3—figure supplement 2*).

Even though patients carry the eIF2B mutation(s) in all cell types, VWM manifests as a CNS disease with the exception of ovarian failure in late-onset female patients. In extremely rare and severe cases, renal dysplasia and hepatomegaly have also been recently reported (*Hamilton et al., 2018*). To evaluate the impact of the R191H mutation on other tissues, we interrogated ISR target expression in various peripheral organs. Upregulation of ATF4 targets was not detected in skeletal muscle, liver, kidney or ovaries (*Figure 3—figure supplement 3*), demonstrating that the CNS is particularly sensitive to a reduction in eIF2B function.

## 2BAct normalized the R191H brain transcriptome

Because our targeted RNA panel consisted of only 15 genes, we turned to RNA-seq in order to comprehensively profile the transcriptional changes that take place in the VWM brain. We analyzed cerebellum from WT and R191H mice at 2, 5 and 7 months of age in order to assess potential changes in

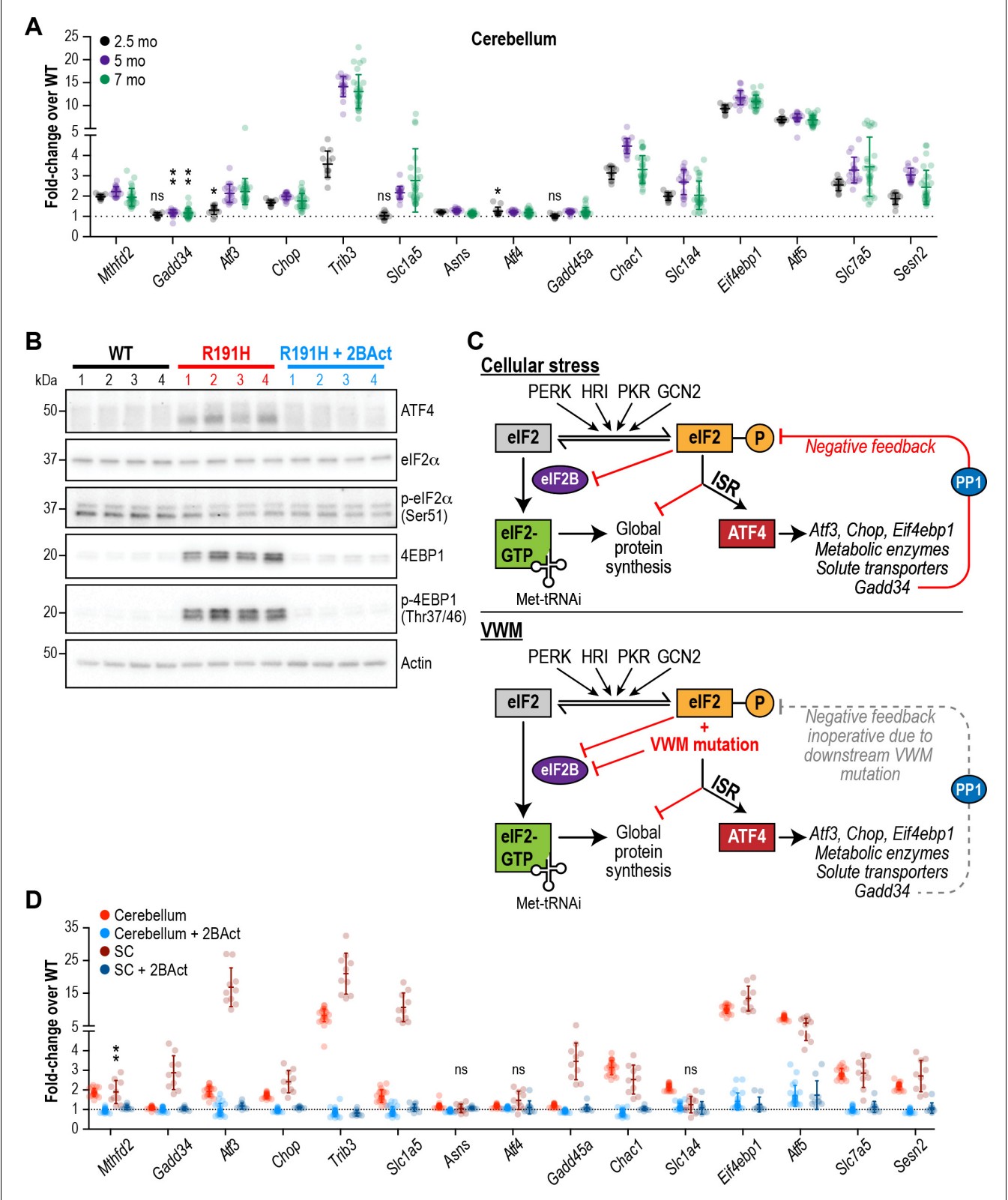

**Figure 3.** The ISR is activated in the brain of R191H mice and its induction is prevented by 2BAct. (**A**) mRNA expression in R191H cerebellum at 2.5 (N = 13/genotype), 5 (N = 20 WT, 19 R191H) and 7 (N = 30/genotype) months of age. (**B**) Western Blots of the indicated proteins from 7-month-old male cerebellum. Actin was used as a loading control. Each lane represents an individual animal. (**C**) Schematic of ISR activation in the context of external stressors or VWM. PP1, protein phosphatase 1. Gadd34, an ATF4-induced regulatory subunit of PP1 that targets it to eIF2. (**D**) mRNA

*Figure 3 continued on next page*

*Figure 3 continued*

expression in R191H cerebellum (N = 23 WT, 21 R191H, 24 R191H + 2 BAct) and spinal cord (N = 10/condition) at 27–32 weeks of age from the 2BAct treatment study (**Figure 1B**). For (**A**) and (**D**), males and females were combined as there was no significant difference between sexes. Data are shown normalized to WT transcript levels. Bars, mean ±SD. *p<0.01; **p<10$^{-3;}$ $^{ns}$p>0.05 by Student's t-test with Holm-Sidak correction (compared to WT). Transcripts without symbols were highly significant with p<10$^{-4}$. 2BAct treatment was highly significant for all transcripts (p<0.01 vs. placebo treatment). A table of p-values from tests is available in **Figure 3—source data 1**.
DOI: https://doi.org/10.7554/eLife.42940.011

The following source data and figure supplements are available for figure 3:

**Source data 1.** Adjusted p-values from t-tests of multiplex transcript expression quantification.
DOI: https://doi.org/10.7554/eLife.42940.015

**Figure supplement 1.** A robust and chronic ISR is triggered in all regions of the R191H mouse brain by 2.5 months.
DOI: https://doi.org/10.7554/eLife.42940.012

**Figure supplement 2.** 2BAct prevents ISR induction in the cerebellum and spinal cord of *Eif2b5*$^{R132H/R132H}$ mice.
DOI: https://doi.org/10.7554/eLife.42940.013

**Figure supplement 3.** The ISR is not induced in peripheral organs of R191H mice.
DOI: https://doi.org/10.7554/eLife.42940.014

R191H mice as they develop pathology. We confirmed the upregulation of ISR target genes beginning as early as 2 months of age, the magnitude of which was sustained at 5 and 7 months (**Figure 4—figure supplement 1A–C**). By contrast, and as expected for a disease driven by dysfunction in eIF2B, we did not observe changes in expression for known downstream targets of the parallel IRE1α or ATF6-dependent branches of the unfolded protein response at any time point (**Figure 4—figure supplement 1A–C**).

In order to identify additional classes of transcripts that distinguish R191H from WT, we performed singular value decomposition (SVD) analysis on the dataset from 2-month-old mice. We focused on genes with the largest positive and negative loadings on the first eigengene (i.e. the first singular vector from SVD [**Alter et al., 2000**]), which separated the samples by genotype (**Figure 4A–B**). The first class consisted of 473 genes with increased expression (positive loadings) in R191H mice. Unsurprisingly, GO-term enrichment analysis revealed categories that contained many known ATF4-dependent targets involved in amino acid metabolism and tRNA aminoacylation (**Supplementary file 1C**) (**Adamson et al., 2016**; **Han et al., 2013**).

A second class comprised 600 genes with reduced expression (negative loadings). These genes were not restricted to expression in a specific cell type, but GO-term enrichment analysis revealed categories related to myelination and lipid metabolism (**Supplementary file 1C**), suggesting an effect on glial cells such as astrocytes and oligodendrocytes. A gene signature of perturbed myelin maintenance is detectable as early as 2 months, preceding evidence of myelin loss by histological analysis. A heatmap of the 50 genes with the largest absolute loadings on the first eigengene revealed that the expression of both classes persisted and was consistent as the animals aged (**Figure 4C**).

Our targeted analysis revealed complete abrogation of the ISR by 2BAct treatment (**Figure 3D**). To test whether 2BAct could also normalize the broad downregulation of transcripts related to CNS function, we performed RNA-seq on cerebellum from 2.5-month-old WT and R191H mice treated for only 4 weeks. Remarkably, both upregulated and downregulated classes of transcripts were normalized in 2BAct-treated R191H mice (**Figure 4D**). Clustering of samples based on the top 50 genes from the previous analysis confirmed that 2BAct-treated R191H mice were indistinguishable from WT (**Figure 4E**). Thus, 2BAct normalized the defective expression of glial and myelination-related genes. Moreover, 2BAct treatment of WT mice did not significantly alter gene expression compared to placebo (**Figure 4—figure supplement 2A–B**). Together, these data demonstrated that 2BAct treatment normalizes the aberrant transcriptional landscape of VWM mice without eliciting spurious gene expression changes in WT mice.

## The ISR is activated in astrocytes and myelinating oligodendrocytes of R191H mice

The robust induction of ISR targets in the brain of VWM mice raised the question of whether all CNS cell types are uniformly affected, or whether a subpopulation of cells is particularly susceptible to a

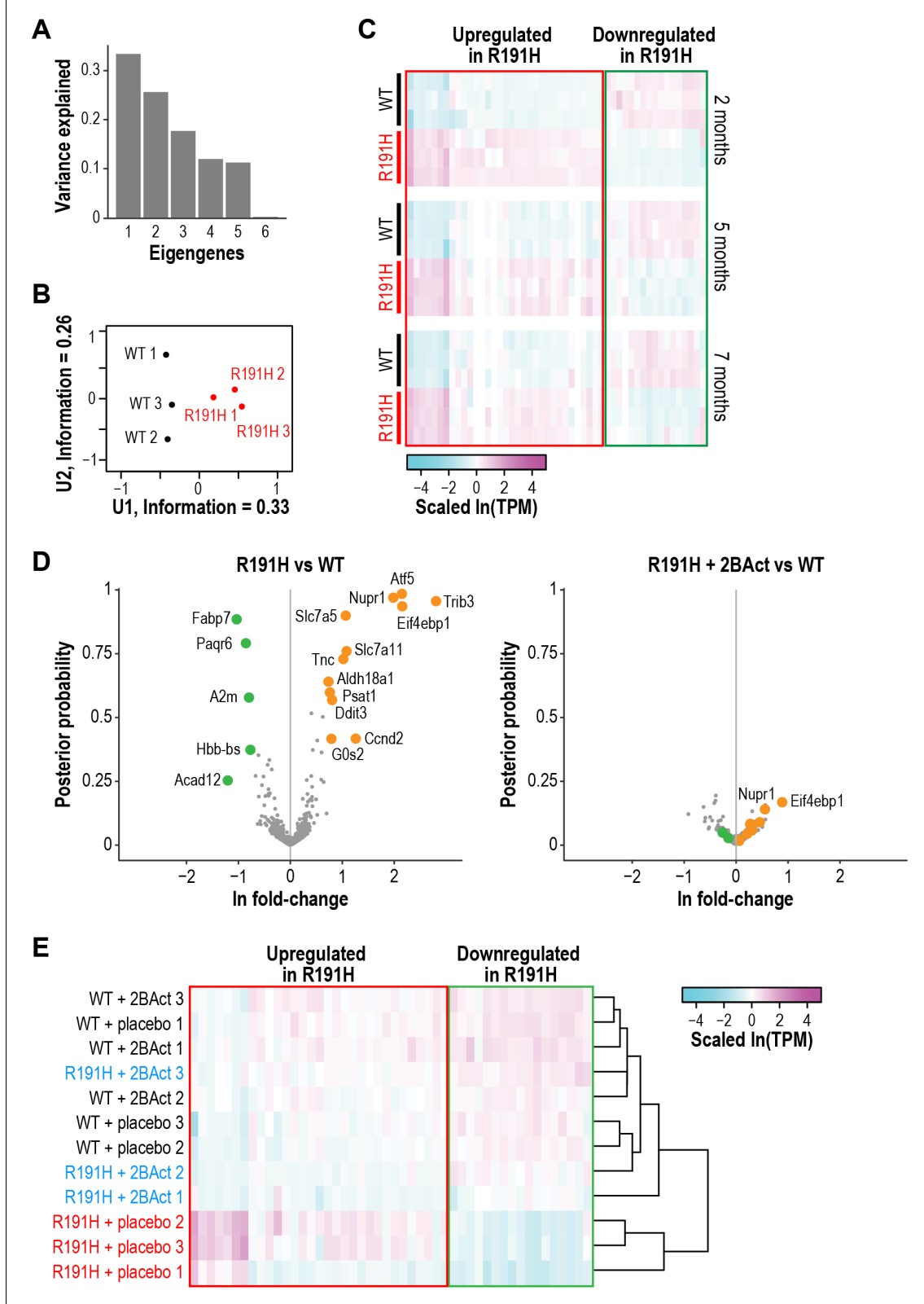

**Figure 4.** R191H mice have an abnormal brain transcriptome at 2 months of age that is normalized by 2BAct treatment. (**A**) Scree plot showing the variance explained by each component of the SVD analysis of 2-month-old WT and R191H cerebellum. (**B**) Individual 2-month-old cerebellum samples plotted along the first and second components of SVD analysis. (**C**) Heatmap of gene expression changes in WT and R191H cerebellum at 2, 5, and 7 months of age (N = 3/genotype/time point). Shown are the 50 genes with the largest absolute loadings in the first eigengene from SVD analysis of 2

*Figure 4 continued on next page*

*Figure 4 continued*

month samples. Source data for (**A**) -(**C**) are available in *Figure 4—source data 1*. (**D**) Volcano plots showing gene expression changes between R191H and WT (*left*) and R191H + 2 BAct and WT (*right*). Orange and green dots indicate transcripts that were more than 2X increased or decreased, respectively, in the R191H vs. WT plot. These dots are replicated on the R191H + 2 BAct vs. WT plot for comparison. (**E**) Heatmap of gene expression changes in WT and R191H cerebellum treated with placebo or 2BAct for 4 weeks. Genes are the same set plotted in *Figure 4C*. Colors indicate the scaled ln(TPM) from the mean abundance of the gene across all samples. For (**D**) and (**E**), N = 3/condition. Source data for (**D**) and (**E**) are available in *Figure 4—source data 2*.

DOI: https://doi.org/10.7554/eLife.42940.016

The following source data and figure supplements are available for figure 4:

**Source data 1.** Fold-changes of transcripts identified in RNA-seq of 2-, 5- and 7-month-old cerebellum.
DOI: https://doi.org/10.7554/eLife.42940.019
**Source data 2.** Fold-changes of transcripts identified in RNA-seq in the 4-week 2BAct treatment experiment.
DOI: https://doi.org/10.7554/eLife.42940.020
**Figure supplement 1.** Sustained ISR induction is a feature of R191H cerebellum across different ages.
DOI: https://doi.org/10.7554/eLife.42940.017
**Figure supplement 2.** 2BAct does not elicit spurious gene changes in WT mice.
DOI: https://doi.org/10.7554/eLife.42940.018

decrease in eIF2B function. To address this, we performed single cell RNA-seq (scRNA-seq) on two brain regions, forebrain and cerebellum, of 2.5-month-old WT and R191H mice. Unbiased clustering of single cells from each region identified 13 and 10 clusters in the forebrain and cerebellum, respectively, that were subsequently assigned cell type identities using CNS cell type gene markers obtained from bulk RNA-seq data (*Figure 5A*, *Figure 5—figure supplement 1* and *Figure 5—figure supplement 2A*) (*Koirala and Corfas, 2010*; *Zhang et al., 2014*). Cells from both WT and R191H tissues were represented in each cluster, demonstrating that transcriptionally defined cell types are not influenced by genotype at this early time point in disease progression. Next, we generated an unbiased, tissue-independent ISR target signature by using the top 50 upregulated genes from our bulk RNA-seq analysis of R191H cerebellum as input for Clustering by Inferred Co-Expression (CLIC) analysis (*Li et al., 2017*). CLIC identified 18 of our input genes as coherently co-expressed across 1774 diverse mouse microarray datasets, and expanded this co-expression module to a final signature comprising 95 genes (*Supplementary file 1D*).

Using the CLIC-derived signature, we first assessed ISR expression in WT cells only. Strikingly, two astrocyte clusters in the forebrain and Bergmann glia in the cerebellum showed significant enrichment of the ISR signature (q = 0.004, 0.008 and 0.004, respectively), indicating that these cell types have higher basal expression of ISR targets (*Figure 5B* and *Figure 5—figure supplement 2B*). By contrast, the ISR signature was insignificant (using a threshold of q = 0.05) or even negatively enriched in other WT CNS cell types.

Next, we investigated the source of upregulated ISR expression in R191H compared to WT. In the forebrain, the two astrocyte clusters with an enriched ISR signature in WT showed significant further upregulation in R191H tissue (q < $10^{-3}$ for both clusters; *Figure 5C*). In the cerebellum, Bergmann glia also showed the most significant enrichment of the ISR signature in R191H tissue (q = 0.0004; *Figure 5—figure supplement 2C*). Astrocytes, and more recently Bergmann glia, have long been implicated as the affected cell type in VWM based on histology and ex vivo analyses (*Bugiani et al., 2011*; *Dietrich et al., 2005*; *Dooves et al., 2016*; *Dooves et al., 2018*; *Geva et al., 2010*; *Leferink et al., 2018*; *Wong et al., 2000*). For the first time, our work provides in vivo evidence that in VWM mice, astrocytes exhibit a molecular signature of an early maladaptive ISR.

Interestingly, the comparison of R191H to WT cerebellum also revealed significant enrichment of the ISR signature in other non-neuronal cell types, including myelinating oligodendrocytes, an unassigned cell type, and endothelial cells (q = 0.001, 0.005 and 0.02, respectively; *Figure 5—figure supplement 2C*). We were unable to confidently assign an identity to the unknown cell type, but its expression profile is consistent with a non-neuronal lineage (cluster four in *Figure 5—figure supplement 1B*). Our findings held true when we repeated the analysis using a manually curated list of ISR target genes (*Figure 5—figure supplement 3* and *Supplementary file 1D*). Collectively, the results of our unbiased analysis suggest that in R191H brain, other cell types beyond astrocytes upregulate the ISR and may contribute to pathology.

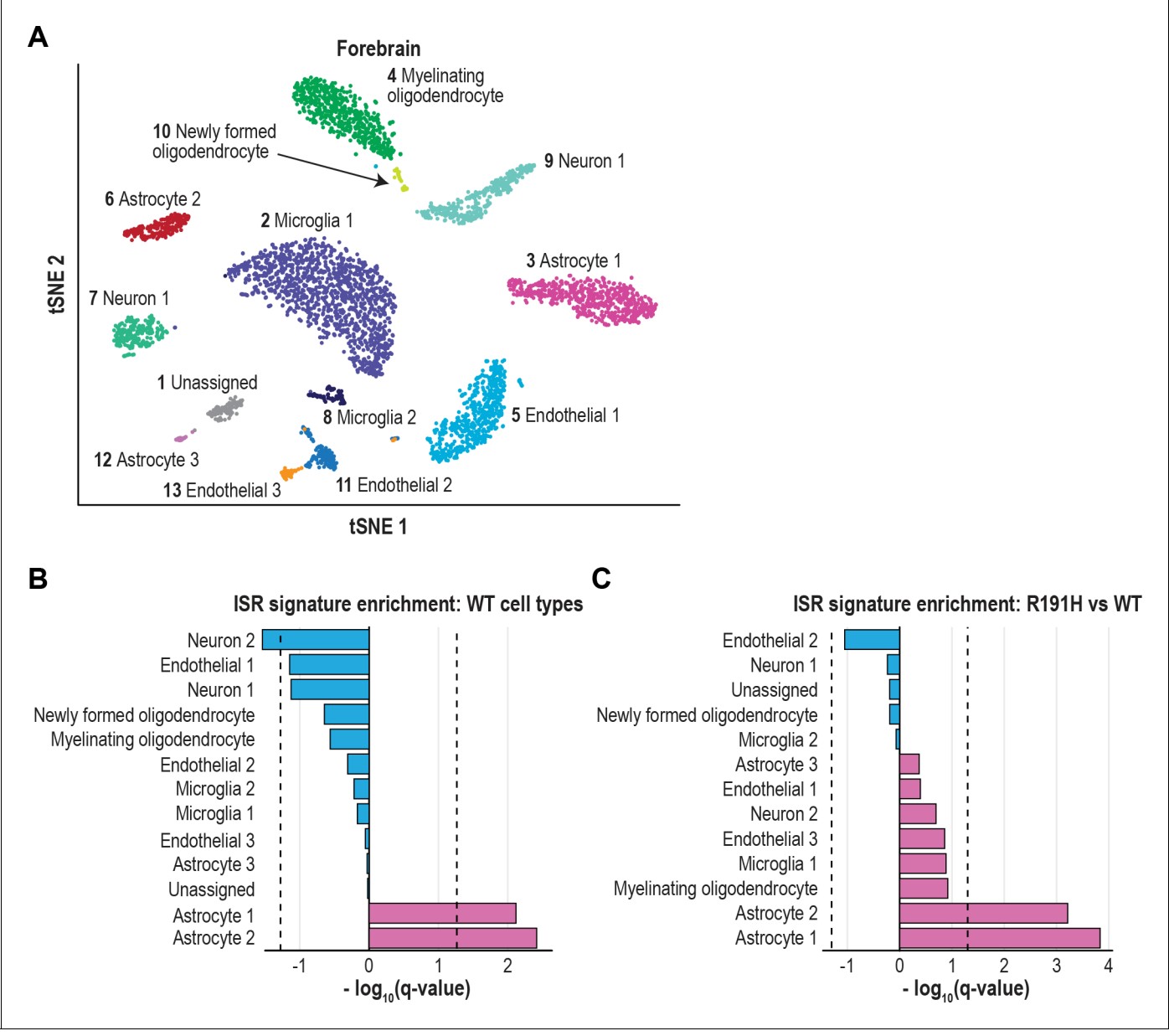

**Figure 5.** The ISR is strongly activated in astrocytes of VWM forebrain. (**A**) tSNE plot showing the 13 transcriptionally defined clusters identified from single-cell analysis of WT and R191H forebrain. (**B**) Q-values from GSEA based on the differential expression analysis of each WT cluster versus all other WT clusters, using the ISR gene expression signature derived from CLIC. (**C**) Q-values from the differential expression analysis of R191H vs WT cells for each cluster, using the same ISR signature as in (**B**). In (**B**) and (**C**), dotted lines indicate Q-value thresholds of 0.05.

DOI: https://doi.org/10.7554/eLife.42940.021

The following figure supplements are available for figure 5:

**Figure supplement 1.** scRNA-seq of WT and R191H forebrain and cerebellum yields distinct transcriptionally defined clusters that do not depend on genotype.

DOI: https://doi.org/10.7554/eLife.42940.022

**Figure supplement 2.** The ISR is strongly activated in Bergmann glia of VWM cerebellum.

DOI: https://doi.org/10.7554/eLife.42940.023

**Figure supplement 3.** A curated list of ISR targets reveals activation in astrocytes and Bergmann glia of VWM brain.

DOI: https://doi.org/10.7554/eLife.42940.024

## 2BAct normalized the R191H brain proteome without rescuing eIF2B levels

Because eIF2B is essential for translation initiation and the *Eif2b5*[R191H] mutation reduces its GEF activity, we wondered how well the changes observed by RNA-seq would correlate with changes in the proteome. To address this, we performed tandem mass tag mass spectrometry (TMT-MS) on cerebellum samples at the end of the 2BAct treatment study.

We discovered 42 proteins that increased >1.5 fold in abundance, and 19 proteins that decreased >1.5 fold in abundance in placebo-treated R191H vs. WT at a posterior probability >90% (*Figure 6A*). Of the upregulated proteins, 21/42 were transcriptionally upregulated in the RNA-seq experiment. The remaining half did not meet an abundance threshold in the RNA-seq analysis. Among these were known ISR targets such as amino acid transporters (SLC1A4 and SLC7A3) and metabolic enzymes (CTH and PYCR1). However, TMT-MS did not detect ATF4 or some of its low-abundance targets (e.g. the transcription factors ATF3, ATF5 and CHOP). Nevertheless, the presence of the other ISR targets in this set drove the enrichment of ISR-associated pathways in GO-term analysis (*Figure 6B*). The good agreement between RNA-seq and TMT-MS data, as well as the ability to detect targets present in one but not the other, highlight their utility as complementary approaches.

Of the downregulated proteins, 13/19 were transcriptionally downregulated in the RNA-seq experiment, one did not change transcriptionally and five did not meet the RNA-seq abundance threshold. We did not identify significant enrichment of gene sets in the downregulated proteins, likely due to the small number of targets that met our cutoff criteria. Remarkably, 14/19 of these targets are most highly expressed in Bergmann glia or astrocytes, and 9/19 are highly expressed in oligodendroyctes or oligodendrocyte precursor cells (*Tabula Muris Consortium et al., 2018*). One interesting example that falls into both categories is the fatty-acid binding protein FABP7, the downregulation of which is again suggestive of dysregulation of lipid metabolism (*Kipp et al., 2011*; *Kurtz et al., 1994*). Our proteomic data are consistent with the downregulated GO categories observed in RNA-seq, as well as the ISR activation seen in astrocytes and myelinating oligodendrocytes by scRNA-seq. Together, they implicate glial cells as a potential source of dysregulation. Importantly, all downregulated proteins and 40/42 upregulated proteins were normalized by 2BAct treatment (*Figure 6A right panel* and *Figure 6C*).

Interestingly, we discovered that levels of all five eIF2B subunits were reduced 15–35% in R191H cerebellum compared to WT (*Figure 6D*). This decrease occurred at the protein level, as no changes were observed in transcript abundance by RNA-seq. We had previously observed destabilization of eIF2B in HEK293T cells, wherein a VWM mutation in one member of the complex caused a reduction in itself as well as the other subunits (*Wong et al., 2018*). The finding that long-term treatment with 2BAct does not rescue eIF2B complex levels in vivo suggests that the normalization of all measured endpoints in R191H mice (*Table 1*) is due to boosting the GEF activity of the remaining mutant complex to functionally normal levels.

## Discussion

Introduction of a severe VWM mutation into mice led to spontaneous loss of myelin and motor deficits that reproduce key aspects of the human disease. The impaired GEF activity of mutant eIF2B underlies the observed translational upregulation of the transcription factor ATF4 and in turn, induction of a maladaptive ISR program that precedes behavioral pathology. Our data demonstrate that a subset of astrocytes possess a basal ISR in normal mice, which is further activated when eIF2B is mutated. Notably, we also detected ISR upregulation in other non-neuronal cell populations, including myelinating oligodendrocytes. Histological analysis has shown that white matter astrocytes, but not grey matter astrocytes, are dysmorphic and a subset of oligodendrocytes are characterized as foamy in VWM patients (*Hata et al., 2014*; *Wong et al., 2000*). Moreover, Bergman glia, astrocytes found in the cerebellum, and Muller glia, astrocytes found in the retina, were shown to be severely affected in patients and in VWM mouse models (*Dooves et al., 2016*; *Dooves et al., 2018*). In agreement with histology, scRNA-seq of two brain regions identified astrocyte subtypes with exacerbated ISR induction in R191H. As cells in the brain are highly interconnected, cellular and metabolic dysfunction in astrocytes could lead to malfunction in other cell types such as the myelinating

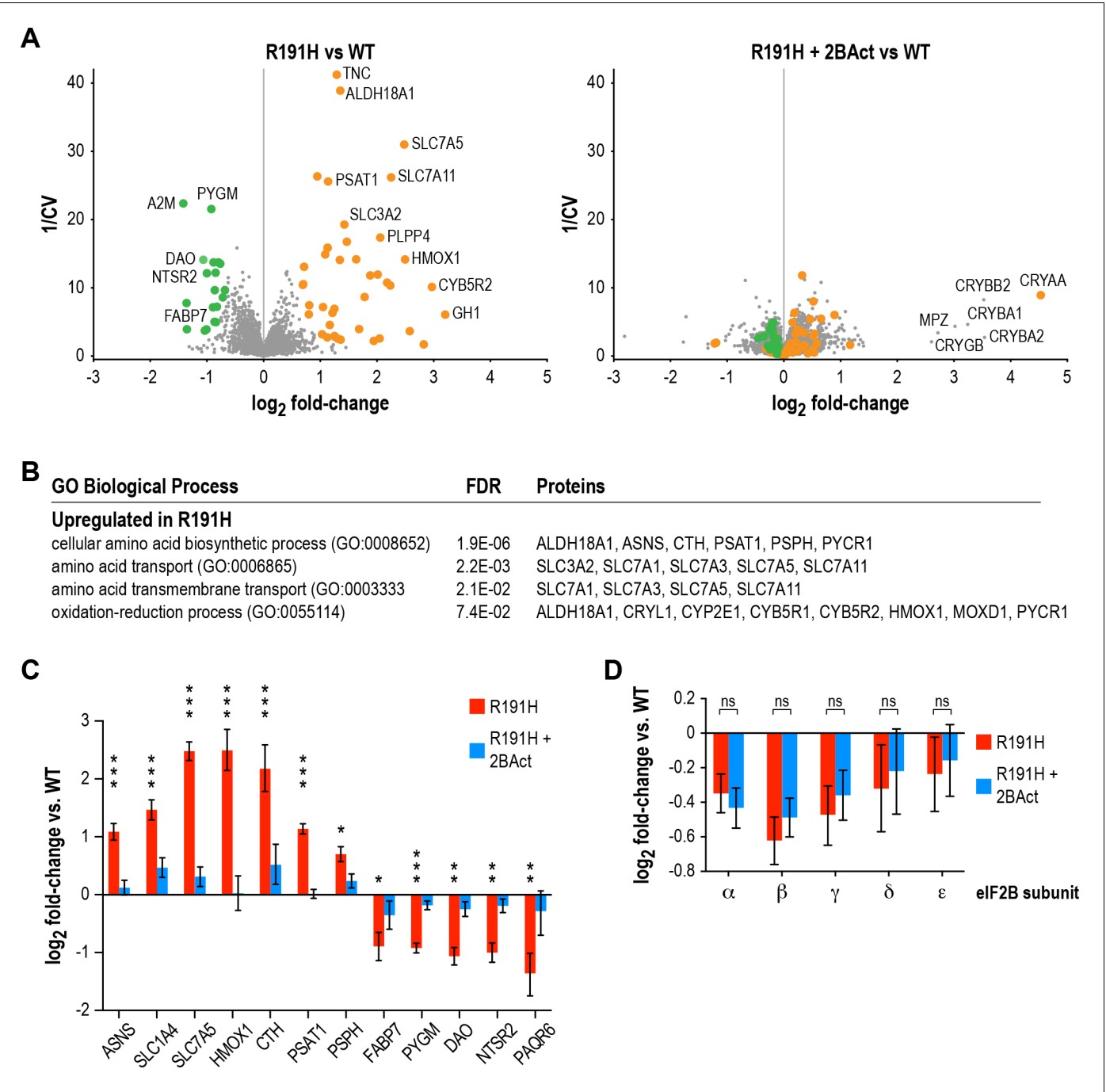

**Figure 6.** 2BAct normalizes the R191H brain proteome without affecting eIF2B subunit levels. (**A**) Volcano plots showing protein abundance changes between R191H and WT (*left*) and R191H + 2 BAct and WT (*right*). The y-axis is the inverse of the coefficient of variation. Orange and green dots indicate proteins that were more than 1.5X increased or decreased, respectively, in the R191H vs. WT plot at a posterior probability of >90%. These dots are replicated on the R191H + 2 BAct vs. WT plot for comparison. (**B**) GO-term enrichment analysis of proteins meeting the threshold for increase in (**A**). Categories shown have an FDR cutoff smaller than $10^{-2}$. Downregulated proteins did not show enrichment for any categories. (**C**) Quantification of selected ISR, metabolic and neural targets relative to WT levels, showing rescue by 2BAct treatment. Posterior probability *<0.05; **<$10^{-5}$; ***<$10^{-10}$ of a <50% difference compared to WT. All targets in the 2BAct-treated condition had posterior probability >0.5 of a <50% difference compared to WT. (**D**) Quantification of eIF2B subunits relative to WT levels. Posterior probability >0.95 of a <25% difference between placebo-treated and 2BAct-treated conditions. For all panels, N = 6/condition. For (**B**) and (**C**), bars are mean ±95% credible intervals. Source data are available in *Figure 6—source data 1*.

DOI: https://doi.org/10.7554/eLife.42940.025

*Figure 6 continued*

The following source data is available for figure 6:

**Source data 1.** Fold-changes of proteins identified in TMT-MS proteomics experiment.
DOI: https://doi.org/10.7554/eLife.42940.026

oligodendrocytes. Whether ISR activation in the non-astrocytic cells is cell-autonomous or triggered by signals from astrocytes remains to be explored.

Among the ATF4 targets revealed by both transcriptomics and proteomics are solute transporters (SLC1A4, SLC1A5, SLC3A2, SLC7A1, SLC7A3, SLC7A5, SLC7A11) and metabolic enzymes (ASNS, CTH, CBS, PLPP4, PHGDH, PSAT1, PSPH, SHMT2 and MTHFD2), and their upregulation is likely to disrupt cellular functions in the CNS. Interestingly, ATF4 is chronically induced in various mouse models of mitochondrial dysfunction, and mutations in human mitochondrial proteins can lead to loss of myelin or leukoencephalopathies (*Carvalho, 2013*; *Dogan et al., 2014*; *Huang et al., 2013*; *Mendes et al., 2018*; *Moisoi et al., 2009*; *Pereira et al., 2017*; *Quirós et al., 2017*; *Taylor et al., 2014*). An important question for future investigation is whether these diseases are also characterized by a chronic ISR that may be protective or maladaptive, and which cell types are susceptible to its induction.

The induction of *Eif4ebp1* and *Sesn2*, two negative regulators of the mTOR signaling pathway, suggests that impairment of translation in VWM may extend beyond the direct effect of crippled eIF2B activity (*Budanov and Karin, 2008*; *Wolfson et al., 2016*; *Yanagiya et al., 2012*). We speculate that inhibition of the eIF2 and mTOR pathways could lead to a dual brake on protein synthesis. If protein synthesis is globally reduced by chronic ISR activation in oligodendrocytes, this could directly disrupt maintenance of the myelin sheath.

The ISR signature of VWM mice is unique as it originates downstream of the canonical sensor, that is eIF2α phosphorylation by stress-responsive kinases. Thus, it provides insight into the transcriptional and proteomic changes driven by the response in vivo in the absence of exogenously applied pleiotropic stressors. In contrast to ISR induction via stress-responsive kinases, the negative feedback loop elicited by both the constitutive CREP-containing and the ISR-inducible GADD34-

**Table 1.** Measured parameters in R191H mice and effect of 2BAct.

| Parameter | R191H phenotype | Effect of 2BAct |
|---|---|---|
| **Physiological** | | |
| Body weight | Reduced | Normalized |
| Inverted grid test | Reduced hang time | Normalized |
| Balance beam test | Longer crossing time, more errors | Normalized |
| **Histological** | | |
| Myelin levels | Reduced | Normalized |
| GFAP staining | Increased | Normalized |
| Iba-1 staining | Increased | Normalized |
| ATF3 staining | Increased | Normalized |
| Olig2 staining | Increased | Normalized |
| **Molecular** | | |
| ATF4 expression | Increased | Normalized |
| ISR target genes expression | Increased | Normalized |
| Transcriptome | Deregulated | Normalized |
| Proteome | Deregulated | Normalized |
| eIF2B complex levels | Reduced | No effect |
| eIF2B specific activity | Reduced | Increased |

DOI: https://doi.org/10.7554/eLife.42940.027

containing eIF2α phosphatases is not effective in attenuating the response in VWM. Therefore, it constitutes a 'locked-on' ISR (*Figure 3C*). However, the system can still respond to further stress, as seen in VWM patients wherein provoking factors (e.g. febrile infections or head trauma) exacerbate the disease and lead to poorer outcomes (*Hamilton et al., 2018*).

2BAct had a normalizing effect on the transcriptome, proteome, myelin content, microglial activation, body weight and motor function of R191H VWM animals. Long-term administration of 2BAct starting at ~8 weeks of age is a preventative treatment paradigm. Beginning treatment at later time points would likely result in ISR attenuation, as we predict that eIF2B activation by 2BAct is age-independent. However, the degree of pathological amelioration in a therapeutic mode would likely depend on the extent of neuronal damage and myelin perturbation in the nervous system at the start of treatment. The accrued damage in turn would depend on the severity of the eIF2B mutant allele and the time elapsed since disease onset.

The remyelination capacity of humans and mice is likely to differ significantly, thus the therapeutic value of eIF2B activators in VWM is best interrogated in the clinic. The small molecule 2BAct has cardiovascular liabilities at a minimal efficacious dose and is not suitable for human dosing. Therefore, therapeutic testing of this class of molecules awaits the generation of a suitable candidate. Nevertheless, ISRIB has shown beneficial effects in mouse models when administered either acutely or for short periods of time by intraperitoneal delivery (*Chou et al., 2017*; *Halliday et al., 2015*; *Li et al., 2018*; *Nguyen et al., 2018*; *Sidrauski et al., 2013*; *Wang et al., 2018*). The improved in vivo properties of 2BAct make it an ideal molecule to interrogate the efficacy of this mechanism of action in rodent disease models that require long-term dosing, including those characterized by chronic activation of any of the four eIF2α kinases.

We and others previously demonstrated that stabilizing the decameric complex can boost the intrinsic GEF activity of both WT and various VWM mutant eIF2B complexes (*Tsai et al., 2018*; *Wong et al., 2018*). To date, all tested VWM mutations have responded to eIF2B activators in vitro. Here, we show that increased GEF activity can compensate in vivo under conditions where a severe mutation leads to both decreased levels of eIF2B complex and reduced intrinsic activity. Furthermore, 2BAct also shut off the ISR in vivo in mice bearing an *Eif2b5*^R132H/R132H VWM mutation. Although it is possible to introduce artificial mutations into eIF2B that interfere with the compound binding site, no VWM mutations are known to exist in this region (*Sekine et al., 2015*; *Tsai et al., 2018*; *Wong et al., 2018*). Based on this, we anticipate that 2BAct-like compounds may be broadly efficacious across the range of mutations identified in the VWM patient population. Finally, our results demonstrate that eIF2B activation is a sound strategy for shutting off the ISR in vivo. This raises the possibility that various diseases exhibiting a maladaptive ISR could be responsive to this mechanism of action.

## Materials and methods

### Key resources table

| Reagent type (species) or resource | Designation | Source or reference | Identifiers | Additional information |
|---|---|---|---|---|
| Genetic reagent (*M. musculus*) | R191H VWM mouse model | this paper | | *Eif2b5*^R191H/R191H mutation in C57BL/6J background |
| Genetic reagent (*M. musculus*) | R132H VWM mouse model | this paper | | *Eif2b5*^R132H/R132H mutation in C57BL/6J background |
| Chemical compound, drug | 2BAct | this paper | | Synthesized in-house |
| Cell line (*H.sapiens*) | HEK293T with ATF4-Luc reporter | PMID: 23741617 | | |
| Commercial assay or kit | Quantigene Plex 2.0 assay | Thermo Fisher Scientific | | Custom gene panel |

*Continued on next page*

*Continued*

| Reagent type (species) or resource | Designation | Source or reference | Identifiers | Additional information |
|---|---|---|---|---|
| Commercial assay or kit | ONE-GLO luciferase assay system | Promega | #E6120 | |
| Antibody | Rabbit polyclonal anti-MBP | abcam | #ab40390 (RRID:AB_11141521) | IHC 5 ug/ml; epitope retrieval with pepsin pH 2.3, 10–20 min |
| Antibody | Mouse anti-GFAP | Millipore | #MAB3402 (RRID:AB_94844) | IHC 1 ug/ml; epitope retrieval with citrate pH 6, 95C, 30 min |
| Antibody | Rabbit polyclonal anti-Iba1 | Wako Chemicals | #019–19741 (RRID:AB_839504) | IHC 1 ug/ml; epitope retrieval with citrate pH 6, 95C, 30 min |
| Antibody | Rabbit monoclonal anti-ATF3 | abcam | #ab207434 (RRID:AB_2734728) | IHC 4 ug/ml; epitope retrieval with EDTA pH 9, 95C, 30 min |
| Antibody | Rabbit monoclonal anti-Olig2 | abcam | #ab109186 (RRID:AB_10861310) | IHC 0.3 ug/ml; epitope retrieval with EDTA pH 9, 95C, 30 min |
| Antibody | Rabbit monoclonal anti-ATF4 | Cell Signaling Technology | #11815 (RRID:AB_2616025) | Western Blot (1:1000 dilution) |
| Antibody | Rabbit polyclonal anti-eIF2α | Cell Signaling Technology | #9722 (RRID:AB_2230924) | Western Blot (1:1000 dilution) |
| Antibody | Rabbit monoclonal anti-phospho-eIF2α | Cell Signaling Technology | #3398 (RRID:AB_2096481) | Western Blot (1:1000 dilution) |
| Antibody | Rabbit monoclonal anti-4EBP1 | Cell Signaling Technology | #9644 (RRID:AB_2097841) | Western Blot (1:1000 dilution) |
| Antibody | Rabbit monoclonal anti-phospho-4EBP1 | Cell Signaling Technology | #2855 (RRID:AB_560835) | Western Blot (1:1000 dilution) |
| Antibody | Mouse monoclonal anti-actin | Cell Signaling Technology | #3700 (RRID:AB_2242334) | Western Blot (1:5000 dilution) |
| Antibody | HRP-conjugated goat anti-rabbit | Promega | #W401B | Western Blot (1:5000 dilution) |
| Antibody | HRP-conjugated goat anti-mouse | Promega | #W402B | Western Blot (1:5000 dilution) |
| Other | Luxol Fast Blue | Electron Microscopy Sciences | #26516–01 | Histological stain for myelin |

## Preparation of 2BAct

To a solution of 5-(difluoromethyl)pyrazine-2-carboxylic acid (20.3 g, 117 mmol) and tert-butyl (3-aminobicyclo[1.1.1]pentan-1-yl)carbamate (22.0 g, 111 mmol) in *N,N*-dimethylformamide (400 mL) at ambient temperature was added triethylamine (61.9 mL, 444 mmol). The 2-(3H-[1,2,3] triazolo[4,5-b] pyridin-3-yl)- 1,1,3,3-tetramethylisouronium hexafluorophosphate(V) (44.3 g, 117 mmol) was added portion wise over 60 min and the mixture was allowed to stir at ambient temperature for 23 hr. The mixture was quenched with saturated, aqueous $NH_4Cl$ (75 mL) and water (30 mL) and diluted with

EtOAc (100 mL). The layers were separated and the aqueous layer was extracted with EtOAc (2 $\times$ 20 mL) and $CH_2Cl_2$ (2 $\times$ 30 mL). The combined organic extracts were dried over anhydrous $Na_2SO_4$, filtered, concentrated under reduced pressure. The residue was crystallized from EtOH ($Et_2O$ wash) to give some solid product. The mother liquor was concentrated under reduced pressure and purified via column chromatography ($SiO_2$, 75% EtOAc/heptanes) to give additional solids. All solids were combined to give tert-butyl (3-(5-(difluoromethyl)pyrazine-2-carboxamido)bicyclo[1.1.1] pentan-1-yl)carbamate (38 g, 107 mmol, 97% yield).

To a solution of tert-butyl (3-(5-(difluoromethyl)pyrazine-2-carboxamido)bicyclo[1.1.1]pentan-1-yl) carbamate (38.0 g, 107 mmol) in $CH_2Cl_2$ (350 mL) at 0°C was added trifluoroacetic acid (132 mL, 1720 mmol) dropwise over 1 hr. This mixture was allowed to warm to ambient temperature and was stirred for 2 hr then was concentrated under reduced pressure and azeotroped with toluene to give N-(3-aminobicyclo[1.1.1]pentan-1-yl) −5-(difluoromethyl)pyrazine-2-carboxamide, trifluoroacetic acid (40.0 g, 109 mmol, quantitative yield).

To a solution of N-(3-aminobicyclo[1.1.1]pentan-1-yl)−5-(difluoromethyl) pyrazine-2-carboxamide, trifluoroacetic acid (39.4 g, 107 mmol) and 2-(4-chloro-3-fluorophenoxy)acetic acid (24.1 g, 118 mmol) in N,N-dimethylformamide (400 mL) was added triethylamine (59.7 mL, 428 mmol). The mixture was cooled to 0°C then 2-(3H-[1,2,3] triazolo[4,5-b]pyridin-3-yl)−1,1,3,3-tetramethylisouronium-hexafluorophosphate (V) (HATU, 44.8 g, 118 mmol) was added portionwise over 30 min and the mixture was allowed to stir at ambient temperature for 16 hr. The mixture was quenched with saturated, aqueous $NH_4Cl$ (100 mL) and water (50 mL) and diluted with EtOAc (200 mL). The layers were separated and the aqueous layer was extracted with EtOAc (2 $\times$ 50 mL) and $CH_2Cl_2$ (2 $\times$ 50 mL). The combined organic extracts were dried over anhydrous $Na_2SO_4$, filtered and concentrated under reduced pressure. The solids were crystallized from EtOAc/heptanes to give some product. The mother liquor was concentrated under reduced pressure and purified via column chromatography ($SiO_2$, 75% EtOAc/heptanes) to give additional solids. All solids were combined and recrystallized again using charcoal to remove color (material in boiling EtOAc and hot filtered) to give product which was dried under vacuum at 45°C for 2 days to give 2BAct (39 g, 88 mmol, 83% yield) as a white solid.

## 2BAct microsuspension preparation

An aqueous suspension of 2BAct was prepared by suspending the drug in 0.5% hydroxypropyl methylcellulose (HPMC; Hypromellose 2910, 4000 mPa.s; Spectrum Chemical Manufacturing Corp, NJ) in water. The suspending vehicle was first prepared by adding 5 g of HPMC to 500 mL of miliQ water heated to 60°C. This mixture was allowed to stir until all of HPMC was dispersed. This solution was then transferred to a volumetric flask with two additional rinses of the original container. Sufficient quantity of water was then added to prepare 1 L of vehicle and allowed to stir overnight to obtain a clear suspension. The vehicle was kept refrigerated and allowed to come to room temperature before each use. Fresh vehicle was prepared every month. For preparation of the aqueous suspension of 2BAct, the compound was weighed into an appropriately sized mortar and levigated with a pestle using a small amount of the vehicle. This was then collected into an appropriately sized glass vial, previously marked with a q.s. line. The mortar was rinsed five times, adding each rinse into the glass vial. Additional vehicle was added to the glass vial until q.s. line was reached and entire suspension mixed by vortexing for 10 s.

## 2BAct pharmacokinetics

Six- to eight-week-old CD1 male mice were dosed with 2BAct at 1 mg/kg or 30 mg/kg orally at a dosing volume of 10 mL/kg. For dosing, 2BAct was micronized and suspended in 0.5% hydroxypropyl methylcellulose (HPMC) (see *Microsuspension preparation* above). Blood was drawn into EDTA charged capillary tubes via the tail vein at the following timepoints: 0.25, 0.5, 1, 3, 6, 9, 12 and 24 hr (N = 3 measurements per timepoint, mice bled at each timepoint, and combined in pairs for extraction). Blood was centrifuged at 3000 rpm and plasma harvested. Plasma samples and standards were extracted by protein precipitation with acetonitrile containing internal standards. The supernatant was diluted with 0.1% formic acid in water before injection into an HPLC-MS/MS system for separation and quantitation. The analytes were separated from matrix components using reverse phase chromatography (30 $\times$ 2.1 mm, 5 µm Fortis Pace C18) using gradient elution at a flow

rate of 0.8 mL/min. The tandem mass spectrometry analysis was carried out on SCIEX triple quadrupole mass spectrometer with an electrospray ionization interface, in positive ion mode. Data acquisition and evaluation were performed using Analyst software (SCIEX).

## Preparation of 2BAct in diet

2BAct was administered orally by providing mice with the compound incorporated in rodent meal (2014, Teklad Global 14% Protein Rodent Maintenance Diet; Envigo, WI). For this, the compound was weighed, added to a mortar with small amount of powdered meal, and ground with a pestle until homogenous. This was further mixed with additional powdered meal in HDPE bottles by either geometric mixing with hand agitation or using a Turbula mixer (Glen Mills Inc., NJ) set at 48 rpm for 15 min or contract manufactured at Envigo to achieve a 2BAct concentration of 300 ppm (300 µg 2BAct/g of meal). Teklad 2014 without added compound was offered as the placebo diet.

## Generation of mouse models

The $Eif2b5^{R191H/R191H}$ knock-in mutant mouse model was generated in the background strain C57BL/6J as a service by genOway (Lyon, France). Briefly, a targeting vector was designed against the $Eif2b5$ locus to simultaneously insert: (1) a Flp-excisable neomycin resistance cassette between exons 2 and 3; (2) a CGC - > CAC codon substitution in exon 4 (changing residue Arg191 to His); (3) loxP sites flanking exons 3 and 7 (*Figure 1—figure supplement 2*). Successful homologous recombination in ES cells was verified by PCR and Southern Blotting. Chimeras were generated by blastocyst injection, which were then mated to WT C57BL/6J mice to identify F1 heterozygous $Eif2b5^{+/R191H; FRT-neo\ (flox)}$ progeny. The neomycin resistance cassette was removed by mating of heterozygous mice to Flp deleter mice. The resulting $Eif2b5^{+/R191H\ (flox)}$ mice were used as colony founders. Experiments were performed using homozygous mutant mice and their WT littermates as controls. The $Eif2b5^{R132H/R132H}$ mouse model was generated in a similar manner.

## Mouse embryonic fibroblast isolation

Pregnant female C57BL/6J mice (WT and $Eif2b5^{R191H\ (flox)/R191H\ (flox)}$) were sacrificed 13 days after discovery of a post-mating plug. Embryos were dissected out of the uterine horn and separated for individual processing. The head and blood-containing organs of each embryo were removed. The remaining tissue was finely minced with a razor blade and triturated using a pipet. Cells were dissociated with 0.25% trypsin-EDTA for 30 min at 37°C. Dissociated cells were washed once in DMEM +10% FBS+1X antibiotic-antimycotic solution, then plated onto 10 cm dishes in fresh medium. Cells were expanded for two passages before freezing for storage. Cells from each embryo were treated as separate, independent lines.

## Western blots

Cerebellum lysates were prepared in RIPA buffer (Sigma #R0278)+protease/phosphatase inhibitors (Pierce #A32959). Tissues were lysed in a Qiagen TissueLyser II for 2 × 2 min intervals at 30 Hz. Lysates were incubated on ice for ten minutes and centrifuged (21,000 x g, 10 min, 4°C) to remove cellular debris. Protein concentrations were determined using a Pierce BCA assay (Thermo #23227) and adjusted to 2 mg/mL using RIPA buffer. Lysates were aliquoted, flash-frozen and stored at −80°C. For western blots, samples were run on Mini-PROTEAN TGX 4–20% gradient gels (Bio-Rad #4561096) and transferred using Trans-Blot Turbo Mini-PVDF Transfer packs (Bio-Rad #1704156) on a Trans-Blot Turbo apparatus. Membranes were blocked with 5% milk in TBS-T and incubated overnight with primary antibody in the same blocking buffer at 4°C. After three washes of 15 min each in TBS-T, HRP-conjugated secondary antibodies were applied for 1 hr. Membranes were washed in TBS-T as before. Advansta WesternBright chemiluminescent substrate was applied to the membranes and images were obtained on a Bio-Rad ChemiDoc MP imaging system in signal accumulation mode.

## ATF4-luciferase reporter assay

The experiment was performed as previously described (*Wong et al., 2018*). Briefly, HEK293T cells expressing an ATF4-luciferase reporter (*Sidrauski et al., 2013*) were seeded into 96-well plates and treated with 100 nM thapsigargin for 7 hr to induce ER stress. Cells were co-treated with 2BAct or

ISRIB in dose response. Luminescence was measured using ONE-Glo Luciferase assay reagent (Promega) and a Molecular Devices SpectraMax i3x plate reader. Data were analyzed in Prism (GraphPad Software).

## GEF assay

The experiment was performed as previously described (*Wong et al., 2018*). Briefly, Bodipy-FL-GDP-loaded eIF2 was used as a substrate for lysates generated from WT and R191H MEFs. The assay was performed in 384-well plates. In a final assay volume of 10 µL/well, the following conditions were kept constant: 25 nM Bodipy-FL-GDP-loaded eIF2, 3 nM phospho-eIF2, 0.1 mM GDP, 1 mg/mL BSA, 0.1 mg/mL MEF lysate. 2BAct was dispensed from a 1 mM stock. For each run, triplicate measurements were made for each concentration of 2BAct. Reactions were read on a SpectraMax i3x plate reader using the following instrument parameters: plate temperature = 25°C; excitation wavelength = 485 nm (15 nm width); emission wavelength = 535 nm (25 nm width); read duration = 30 mins at 45 s intervals. Data were analyzed in Prism. GDP release half-lives were calculated by fitting single-exponential decay curves. EC50s were calculated by fitting log(inhibitor) vs response curves.

## Animal care

Mice were allowed to habituate to our facilities for at least 1 week prior to the start of experiments. Mice were housed on a 12-hr light/dark cycle (lights on at 6:00, lights off at 18:00) in a temperature- and humidity-controlled environment (22 ± 1°C, 60–70% humidity). Experimental procedures took place during the illuminated phase of the cycle. Male mice were individually housed due to aggression. Female mice were housed as shipped from the vendor in groups of 1–3 animals/cage. Animals had free access to food and water. AbbVie is committed to the internationally accepted standard of the 3Rs (Reduction, Refinement, Replacement) and adhering to the highest standards of animal welfare in the company's research and development programs. Animal studies were approved by AbbVie's Institutional Animal Care and Use Committee or Ethics Committee. Animal studies were conducted in an AAALAC-accredited program where veterinary care and oversight was provided to ensure appropriate animal care.

Body weights were measured weekly for mice throughout the study. For experiments, mice were pseudo-randomly balanced between treatment groups by date of birth. Analysis of groups was performed in a blinded fashion. Due to the number of animals required for the 2BAct treatment study, mice were binned into age cohorts spanning 5 weeks each. Power analysis was not done prior to beginning the study as experiments were animal-limited. Post-hoc analysis demonstrated that the sample sizes used were sufficient to detect an effect size of 1.1 at 90% power and alpha = 0.05 (two-sided t-test).

## Beam walking

A 100 cm long x 2.5 cm diameter PVC pipe, suspended 30 cm above a padded landing area, was used as the balance beam. A length of 80 cm was marked on the pipe as the distance the animals were required to cross. The pipe was placed in a dimly lit room, with a spotlight suspended above the starting end to serve as the aversive condition. A darkened enclosure with bedding was placed at the opposite end to promote a more agreeable condition. A video camera was set above the starting end to record foot slips as the animal progressed across the pipe.

Animals were habituated to the experimental room for at least one hour prior to testing. To begin an experiment, each subject was placed on the starting end of the beam and allowed a cutoff time of 30 s to cross. Elapsed time was only recorded while the subject actively moved towards the finish line; if it turned around or stopped to groom, the timer was paused. If a fall occurred, the subject was restarted at the beginning of the beam. Each subject was given three attempts to complete the balance beam. If a subject failed all three attempts, it was assigned a time of 30 s and a value of 3 falls. The number of foot slips was quantified from the video recording only if the animal completed the task. The balance beam was cleaned and dried between each subject. Due to the non-normal distribution of the data, they were analyzed using a Mann-Whitney (non-parametric) test in Prism. Reported p-values are adjusted for multiple comparisons by the Bonferroni method.

## Inverted grid

Animals were habituated to the experimental room for 30–60 mins prior to testing. A grid screen measuring 20 cm x 25 cm with a mesh density of 9 squares/cm$^2$ was elevated 45 cm above a cage with bedding. Each subject was placed head oriented downward in the middle of the grid screen. When it was determined that the subject had proper grip on the screen, it was inverted 180°. The hang time (duration a subject held on to the screen without falling) was recorded, up to a cutoff time of 60 s. Any subject that was able to climb onto the top of the screen was assigned a time of 60 s. The grid was cleaned and dried between trial days. Due to the non-normal distribution of the data, they were analyzed using a Mann-Whitney (non-parametric) test in Prism. Reported p-values are adjusted for multiple comparisons by the Bonferroni method.

## Immunohistochemistry

Mice were deeply anesthetized with $CO_2$ and rapidly fixed by transcardiac perfusion with normal saline followed by 10% formalin. Brains and spinal cords were excised and allowed to post-fix in formalin for an additional 2 hr. Samples were processed for paraffin embedding, sectioned at 6 μm, and mounted on adhesive-coated slides. Immunohistochemical staining was carried out using the antibodies described in the Key Resources Table and species-appropriate polymer detection systems (ImmPRESS HRP, Vector Laboratories), and developed with 3,3′-Diaminobenzidine (DAB) followed by hematoxylin counterstaining. For GFAP, DAB development was enhanced with nickel. Myelin staining by Luxol Fast Blue was carried out by the Klüver-Barrera method (*Kluver and Barrera, 1953*). Sections were dehydrated through successive ethanol solutions, cleared in xylene, and coverslipped using xylene-based mounting media.

To examine histopathological endpoints in the brain, sections from two coronal levels of the corpus callosum were examined beginning at approximately Bregma + 0.7 and+1.0 mm. For the spinal cord, coronal sections from cervical and thoracic levels were examined. Image capture and analysis was achieved using either a 3DHISTECH Pannoramic 250 Digital Slide Scanner (20X, Thermo Fisher Scientific) and HALO image analysis software (version 2.1.1637.26, Indica Labs), or a BX-51 microscope fitted with a DP80 camera (10X) and cellSens image analysis software (Olympus). The same parameters for microscopy and image analysis were uniformly applied to all images for each endpoint. For the spinal cord, the mean area fraction from a single section from both the cervical and thoracic levels served as the value for each subject to normalize to the different size of the anatomical levels. For the corpus callosum, the mean area of positive staining from a single section at each coronal level served as the value for each subject. Data were analyzed by one-way (for single time point experiments) or two-way ANOVA (for multiple time point experiments). Reported p-values are adjusted for multiple comparisons by the Holm-Sidak method.

## RNA isolation

Total RNA was isolated from ~20 mg pieces of each frozen tissue, with a minimum of one extraction performed from each sample. All steps were performed on dry ice prior to the addition of 350 μL of RTL Buffer + 40 μM DTT. Samples were homogenized at 4°C using a Qiagen TissueLyser II (Retsch, Castleford, UK) for 2 × 2 min intervals at 30 Hz with the addition of one 5 mm stainless steel bead (Qiagen catalog # 69965). This was followed by incubation in a Vortemp 56 (Labnet International, Edison, NJ) at 65°C with shaking (300 rpm). The samples were then centrifuged at 15,000 x g for 5 min. RNA extraction was performed using the RNeasy Mini kit (Qiagen, Germany) according to manufacturer's instructions, eluting in 30 μL nuclease-free $H_2O$. RNA concentration and purity were determined spectrophotometrically.

## QuantiGene plex 2.0 assay

To measure multiplexed transcript expression levels, we designed a custom QuantiGene Plex Panel (*Flagella et al., 2006*) (Thermo Fisher Scientific, Waltham, MA). Either purified RNA or crude lysates (~15–20 mg of frozen tissue, homogenized using the QuantiGene 2.0 Sample Processing Kit) were used since comparable results were achieved in a head-to-head comparison with a set of samples (data not shown). RNA or tissue homogenates were then subjected to the QuantiGene assay following manufacturer's instructions. Briefly, this involved: (1) capturing target RNAs to corresponding genes on specified beads through an overnight hybridization; (2) a second hybridization to capture

the biotinylated label probes for signal amplification; and (3) a final hybridization to capture the SAPE reagent, which emits a fluorescent signal from each bead set. Signal intensities for each bead set, proportional to the number of captured target RNA molecules, were read on a Luminex Flexmap 3D (Luminex, Northbrook, IL). To control for differences in total RNA input between reactions, background-subtracted output data were normalized to the mean of two housekeeping genes, RPL13A and RPL19. The final data are presented as fold-change for each gene relative to the mean of the corresponding WT samples. Data were analyzed in Prism. Student's t-test was performed for the panel of genes comparing R191H to WT for each time point and brain region. Reported p-values are adjusted for multiple comparisons by the Holm-Sidak method.

## RNA-seq library preparation and sequencing

RNA-seq libraries were prepared using purified RNA isolated as described above. RNA quality and concentration were assayed using a Fragment Analyzer instrument. RNA-seq libraries were prepared using the TruSeq Stranded Total RNA kit paired with the Ribo-Zero rRNA removal kit (Illumina, San Diego, CA). Libraries were sequenced on an Illumina HiSeq 4000 instrument. For the experiment comparing different ages, N = 3 males/genotype/time point. For the 4-week 2BAct treatment experiment, N = 3 females/condition.

## RNA-seq data analysis

RNA-seq library mapping and estimation of expression levels were computed as follows. Reads were mapped with STAR aligner (*Dobin et al., 2013*), version 2.5.3a, to the mm10 reference mouse genome and the Gencode vM12 primary assembly annotation (*Mudge and Harrow, 2015*), to which non-redundant UCSC transcripts were added, using the two-round read-mapping approach. This means that following a first read-mapping round of each library, the splice-junction coordinates reported by STAR, across all libraries, are fed as input to the second round of read mapping. The parameters used in both read-mapping rounds are: outSAMprimaryFlag ='AllBestScore', outFilterMultimapNmax ='10', outFilterMismatchNoverLmax ='0.05', outFilterIntronMotifs ='RemoveNoncanonical'. Following read-mapping, transcript and gene expression levels were estimated using MMSEQ (*Turro et al., 2011*). Transcripts and genes which were minimally distinguishable according to the read data were collapsed using the mmcollapse utility of MMDIFF (*Turro et al., 2014*), and the Fragments Per Kilobase Million (FPKM) expression units were converted to Transcript Per Million (TPM) units.

In order to test for differential expression between the R191H and WT samples, for each of the three time points, we used MMDIFF with a design matrix corresponding to two groups without extraneous variables. In order to test whether R191H relative to the WT fold-change varied between each of the two consecutive time points (i.e., 5 months vs. 2 months, and 7 months vs. 5 months), we used MMDIFF with the following design matrix testing whether the fold-change between groups A and B (e.g: R191H and WT at 5 months) is different from the fold-change between groups C and D (e.g: R191H and WT at 2 months): # M 0, 0, 0, 0, 0, 0; # C 0 0, 0 0, 0 0, 0 1, 0 1, 0 1; # P0 1; # P1 1 0 0, 0 1 0, 0 0 1. Genes were ranked in descending order according to the Bayes factor and the posterior probability, reported by MMDIFF.

Singular value decomposition analyses were carried out using R's SVD function. The right singular vectors of the decomposition, referred to as eigengenes, are used to capture the canonical gene expression patters in the data set (*Alter et al., 2000*). We applied a threshold on the absolute gene loading values defined as two-fold the mean of all absolute values. For GO-term enrichment, analyses were carried out using the online tool at geneontology.org (*Ashburner et al., 2000*; *The Gene Ontology Consortium, 2017*). Samples were clustered using the heatmap.2 function from the gplots package (*R Core Team, 2018*; *Warnes et al., 2005*).

## Preparation of samples for scRNA-seq

Forebrain and cerebellum were dissected from 2-month-old female mice. Tissues were dissociated by incubation in 2 mg/mL papain solution (BrainBits, Springfield, IL) for 30 min at 37°C with agitation, followed by trituration. To remove large debris, suspensions were successively passed through 100 μm and 40 μm strainers. Cells were pelleted by centrifugation at 280 x g, and pellets were subjected to the Miltenyi Debris Removal protocol with Debris Removal solution according to manufacturer's

instructions (Miltenyi Biotec, Bergisch Gladbach, Germany). Following this, cells were re-filtered through a 40 µm Flowmi cell strainer (Belart, Wayne, NJ).

Single cell RNA-seq libraries were created using the Chromium Single Cell 3′ Library and Gel Bead Kit v2 and associated consumables (10X Genomics, Pleasanton, CA).~7000 cells per sample were loaded onto the Chromium microfluidic chip for an expected recovery of 4000 cells per sample. Manufacturer's instructions were followed for library preparation, and cDNAs were amplified for 12 cycles. Samples were sequenced on an Illumina HiSeq 4000.

## scRNA-seq data analysis

scRNA-seq fastq files were demultiplexed to their respective barcodes using the 10X Genomics Cell Ranger mkfastq utility. Unique Molecular Identifier (UMI) counts were generated for each barcode using the Cell Ranger count utility, with the mm10 reference mouse genome used for mapping reads. For each sample, barcodes that were not likely to represent captured cells were filtered out by detecting the first local minimum above two in a distribution of $\log_{10}$(UMIs).

To identify cell clusters, the four samples were concatenated and we defined genes with variable expression dispersion using Seurat (*Butler et al., 2018*), which resulted in 2622 genes. PCA was performed on these genes using the rsvd R package to reduce the dimensionality of the data, retaining the 50 PCs explaining the highest amount of variation. We then used Seurat's methodology to build a Shared Nearest Neighbor (SNN) graph of these cell-embedding data, first generating a K-Nearest Neighbor (KNN) graph using K = min(750,#cells-1) and a Jaccard distance cutoff of 1/15. The SNN graph was then used as input to the Louvain algorithm, implemented in the ModularityOptimizer software (*Waltman and van Eck, 2013*). Since this implementation uses a resolution parameter that strongly affects the number of clusters, we searched the 0.05–1.225 range of this parameter, using the mean unifiability isolability clustering metric as our maximization parameter. This process was initially done on all cells in our data, and subsequently repeated for each cluster individually, in an iterative manner where convergence was defined as not being able to break down a cluster into subclusters. This resulted in 10 and 13 clusters in the cerebellum and forebrain, respectively.

In order to obtain gene markers for each cluster, we ran a differential expression test implemented in Seurat (using a likelihood ratio test between a model that assumes that a gene's expression values of two compared clusters values were sampled from two distributions, versus a null model which assumes they were sampled from a single distribution). A marker gene of a given cluster was defined as a gene which was found to be significantly overexpressed (adjusted p-value<0.05) in that cluster compared to all other clusters.

In order to assign cell identities to our identified clusters, we utilized published bulk RNA-seq gene expression data obtained by cell sorting of major brain cell types (*Koirala and Corfas, 2010*; *Zhang et al., 2014*). For each gene that is a cell-type marker in the list compiled from the published data and is expressed in a cluster, we computed its mean expression level across all cells of the cluster and multiplied it by the fraction of cells it was captured in. We then summed these gene scores across all markers of the cell type that are expressed in the cluster of interest and divide that sum by the number of markers of that cell type to account for possible ascertainment biases (as some cell types may have many more markers than others). Finally, we scaled the cell-type scores of each cluster to sum up to one and hence reflect probabilities. Whenever a probability was higher than 0.5, we assigned the identity of that cluster to be of the high probability cell type.

In order to detect whether ISR activation is cell-type-specific in the WT samples, for each gene in each cluster we performed a differential expression analysis between that cluster and all other clusters as described above. Since the clustering analysis using both WT and R191H samples (described above) found that transcriptionally defined clusters are not influenced by genotype, we used these cluster assignments for the WT, as independent clustering of the WT data alone was not powered enough to obtain the same clustering as when performed on both genotypes. Subsequently, for each cluster we performed a Gene Set Enrichment Analysis (GSEA), using the fgsea R package, with our ISR gene lists – one derived from the CLIC analysis (*Li et al., 2017*) and a manually curated list of ATF4 targets (*Supplementary file 1D*), as the gene sets. In order to detect whether ISR activation is cell type-specific between WT and R191H, for each cluster in the combined WT and R191H data, for each gene in each cluster we performed a differential expression analysis contrasting the cells that correspond to the two genotypes. We subsequently performed the ISR GSEA for each cluster.

## Proteomics

Frozen brain samples were lysed with 0.9 mL 50 mM HEPES, 75 mM NaCl, 3% SDS, pH 8.5 + protease/phosphatase inhibitors and homogenized using a Qiagen TissueLyser II for 8 cycles at 20 Hz. Samples were then sonicated for 5 min. Lysates were centrifuged (16,000 x g, 5 min) to remove cellular debris. Proteins were reduced with 5 mM dithiothreitol (56°C, 30 min) and alkylated with 15 mM iodoacetamide (room temperature, 30 min). Excess iodoacetamide was quenched with 5 mM dithiothreitol (room temperature, 30 min). Proteins were precipitated by sequential addition of 4 volumes methanol, 1 vol chloroform and three volumes water, with vigorous vortexing in between. Protein pellets were washed with methanol, air-dried, and resuspended in 50 mM HEPES, 8 M urea, pH 8.5. The samples were then diluted to 4 M urea and digested with Lys-C protease (25°C, 15 hr; WAKO Chemicals, Richmond, VA). The urea concentration was then diluted to 1 M and the samples were digested with trypsin (37°C, 6 hr; Promega, Madison, WI).

After protein digestion, samples were acidified with 10% trifluoroacetic acid and desalted using C18 solid-phase extraction columns (SepPak). Samples were eluted with 40% acetonitrile/0.5% acetic acid followed by 80% acetonitrile/0.5% acetic acid, and dried overnight under vacuum at 30°C. Peptide concentrations were measured using a Pierce BCA assay, split into 50 μg aliquots, and dried under vacuum.

Dried peptides were resuspended in 50 μL 200 mM HEPES/30% anhydrous acetonitrile. 5 mg TMT reagents (Thermo Fisher Scientific) were dissolved in 250 μL anhydrous acetonitrile, of which 10 μL was added to peptides. TMT reagent 131 was reserved for the 'bridge' sample and the other TMT reagents (126, 127 c, 127 n, 128 c, 128 n, 129 c, 129 n, 130 c and 130 n) were used to label the individual samples. Following incubation at room temperature for 1 hr, the reaction was quenched with 200 mM HEPES/5% hydroxylamine to a final concentration of 0.3% (v/v). TMT-labeled amples were acidified with 50 μL 1% trifluoroacetic acid and pooled into 11-plex TMT samples at equal ratios, desalted with SepPak, and dried under vacuum.

Pooled TMT-labeled peptides were fractionated using high pH RP-HPLC. Samples were resuspended in 5% formic acid/5% acetonitrile and fractionated over a ZORBAX extended C18 column (Agilent, 4.6 mm ID x 250 mm length, 5 μm particles). Peptides were separated on a 75 min linear gradient from 5% to 35% acetonitrile in 10 mM ammonium bicarbonate at a flow rate of 0.5 mL/min on an Agilent 1260 Infinity pump (Agilent Technologies, Waldbronn, Germany). The samples were fractionated into a total of 96 fractions, and then consolidated into 12 as described previously (*Edwards and Haas, 2016*). Samples were dried under vacuum and reconstituted in 5% formic acid/ 4% acetonitrile for LC-MS/MS processing.

Peptides were analyzed on an Orbitrap Fusion Lumos mass spectrometer (Thermo Fisher Scientific) coupled to an Easy-nLC (Thermo Fisher Scientific). Peptides were separated on a microcapillary column (100 μm ID x 25 cm length, filled in-house with Maccel C18 AQ resin, 1.8 μm, 120 A; Sepax Technologies). The total LC-MS run length for each sample was 180 min comprising a 165 min gradient from 6–30% acetonitrile in 0.125% formic acid. The flow rate was 300 nL/min and the column was heated to 60°C. Data-dependent acquisition mode was used for mass spectrometry data collection. A high resolution MS1 scan in the Orbitrap (m/z range = 500–1200, resolution = 60,000; AGC = $5 \times 10^5$; max injection time = 100 ms; RF for S-lens = 30) was collected, from which the top 10 precursors were selected for MS2 analysis followed by MS3 analysis. For MS2 spectra, ions were isolated using a 0.5 m/z window using the mass filter. The MS2 scan was performed in the quadrupole ion trap (CID, AGC = $1 \times 10^4$, normalized collision energy = 30%, max injection time = 35 ms) and the MS3 scan was analyzed in the Orbitrap (HCD, resolution = 60,000; max AGC = $5 \times 10^4$; max injection time = 250 ms; normalized collision energy = 50). For TMT reporter ion quantification, up to six fragment ions from each MS2 spectrum were selected for MS3 analysis using synchronous precursor selection.

## Proteomics data analysis

Mass spectrometry data were processed using an in-house software pipeline as previously described (*Huttlin et al., 2010*). Briefly, raw files were converted to mzXML files and searched against a composite mouse Uniprot database (downloaded on 9 May 2017) containing sequences in forward and reverse orientations using the Sequest algorithm. Database searching matched MS/MS spectra with fully tryptic peptides from this composite dataset with a precursor ion tolerance of 20 ppm and a

product ion tolerance of 0.6 Da. Carbamidomethylation of cysteine residues (+57.02146 Da) and TMT tags of peptide N-termini (+229.162932 Da) were set as static modifications. Oxidation of methionines (+15.99492 Da) was set as a variable modification. Linear discriminant analysis was used to filter peptide spectral matches to a 1% FDR (false discovery rate). Non-unique peptides that matched to multiple proteins were assigned to proteins that contained the largest number of matched redundant peptides sequences.

Quantification of TMT reporter ion intensities was performed by extracting the most intense ion within a 0.003 m/z window at the predicted m/z value for each reporter ion. TMT spectra were used for quantification when the sum of the signal-to-noise for all the reporter ions was greater than 200 and the isolation specificity was greater than 0.75 (*Ting et al., 2011*).

Peptide-level data from each 11-plex was column-normalized to have equivalent geometric means and protein level estimates were obtained by fitting a previously described compositional Bayesian model (*O'Brien et al., 2018*). For each protein, $\log_2$ fold-changes are estimated as the posterior means from the model. Variances and 95% credible intervals from the posterior distributions are also reported.

## Acknowledgements

Mike Sheehan and Renee Sadowski for mouse behavioral testing; Baby Martin-McNulty and David Finkle for help harvesting tissues for MEF isolation and early gene expression analysis; Kevin Wright for advice on statistics; Calvin Jan, Scott McIsaac, David Botstein and David Hendrickson for critical reading.

## Additional information

### Competing interests

Yao Liang Wong, Lauren LeBon, Nimrod D Rubinstein, Swathi Krishnan, Fiona E McAllister, Nicole V Haste, Jonathon J O'Brien, Margaret Roy, Andrea Ireland: employee of Calico Life Sciences LLC at the time the study was conducted and has no other competing financial interests to declare. Ana M Basso, Kathy L Kohlhaas, Arthur L Nikkel, Holly M Robb, Diana L Donnelly-Roberts, Janani Prakash: employee of AbbVie at the time the study was conducted and has no other competing financial interests to declare. Jennifer M Frost, Lei Shi, Michael J Dart: employee of AbbVie and is listed as an inventor on a patent application WO2017193063 describing 2BAct. Kathleen Martin, Carmela Sidrauski: employee of Calico Life Sciences and is listed as an inventor on a patent application WO2017193063 describing 2BAct. The other authors declare that no competing interests exist.

## Funding

| Funder | Author |
|---|---|
| Calico Life Sciences LLC | Yao Liang Wong |
| | Lauren LeBon |
| | Ana M Basso |
| | Kathy L Kohlhaas |
| | Arthur L Nikkel |
| | Holly M Robb |
| | Diana L Donnelly-Roberts |
| | Janani Prakash |
| | Andrew M Swensen |
| | Nimrod D Rubinstein |
| | Swathi Krishnan |
| | Fiona E McAllister |
| | Nicole V Haste |
| | Jonathon J O'Brien |
| | Margaret Roy |
| | Andrea Ireland |
| | Jennifer M Frost |
| | Lei Shi |
| | Stephan Riedmaier |
| | Kathleen Martin |
| | Michael J Dart |
| | Carmela Sidrauski |

The funder had no role in study design, data collection and interpretation.

## Author contributions

Yao Liang Wong, Conceptualization, Data curation, Formal analysis, Validation, Investigation, Visualization, Methodology, Writing—original draft, Writing—review and editing; Lauren LeBon, Data curation, Formal analysis, Validation, Investigation, Visualization, Methodology, Writing—original draft, Writing—review and editing; Ana M Basso, Conceptualization, Formal analysis, Supervision, Methodology, Writing—review and editing; Kathy L Kohlhaas, Diana L Donnelly-Roberts, Conceptualization, Data curation, Formal analysis, Supervision, Validation, Investigation, Methodology, Writing—review and editing; Arthur L Nikkel, Data curation, Formal analysis, Supervision, Validation, Investigation, Methodology, Writing—review and editing; Holly M Robb, Data curation, Formal analysis, Validation, Investigation, Methodology, Writing—review and editing; Janani Prakash, Data curation, Formal analysis, Validation, Investigation, Methodology; Andrew M Swensen, Supervision, Methodology, Writing—review and editing; Nimrod D Rubinstein, Data curation, Software, Formal analysis, Writing—review and editing; Swathi Krishnan, Formal analysis, Validation, Investigation, Methodology; Fiona E McAllister, Margaret Roy, Resources, Supervision, Methodology; Nicole V Haste, Data curation, Validation, Investigation, Methodology; Jonathon J O'Brien, Data curation, Software, Formal analysis; Andrea Ireland, Investigation; Jennifer M Frost, Conceptualization, Supervision, Investigation, Methodology, Writing—original draft; Lei Shi, Validation, Investigation, Methodology; Stephan Riedmaier, Formal analysis, Supervision, Investigation, Methodology; Kathleen Martin, Michael J Dart, Conceptualization, Project administration, Writing—review and editing; Carmela Sidrauski, Conceptualization, Supervision, Funding acquisition, Visualization, Writing—original draft, Project administration, Writing—review and editing

## Author ORCIDs

Yao Liang Wong http://orcid.org/0000-0003-0298-8510
Lauren LeBon http://orcid.org/0000-0003-3205-1948
Jonathon J O'Brien http://orcid.org/0000-0001-9660-4797
Carmela Sidrauski http://orcid.org/0000-0002-4850-3112

## Ethics

Animal experimentation: AbbVie is committed to the internationally-accepted standard of the 3Rs (Reduction, Refinement, Replacement) and adhering to the highest standards of animal welfare in the company's research and development programs. Animal studies were approved by AbbVie's Institutional Animal Care and Use Committee or Ethics Committee. Animal studies were conducted in an AAALAC accredited program where veterinary care and oversight was provided to ensure

appropriate animal care. The protocol IDs associated with the ethical approval of the work are 1401D00001 and 1712D00044.

## Decision letter and Author response

Decision letter https://doi.org/10.7554/eLife.42940.035
Author response https://doi.org/10.7554/eLife.42940.036

## Additional files

### Supplementary files

• Supplementary file 1. (**A**) Binding of 2BAct at 10 µM to the CEREP panel of receptors/enzymes/ion channels. Compound binding was calculated as a % inhibition of the binding of a radioactively labeled ligand specific for each target. Results showing inhibition or stimulation higher than 50% are considered to represent significant effects of the test compound. (**B**) Pharmacokinetic data from 2BAct dosing in mice. (**C**) GO Biological Processes identified by enrichment analysis of RNA-seq data. (**D**) List of genes used for enrichment analysis of scRNA-seq data.
DOI: https://doi.org/10.7554/eLife.42940.028

• Transparent reporting form
DOI: https://doi.org/10.7554/eLife.42940.029

### Data availability

All data generated or analysed during this study are included in the manuscript and supporting files. Source data files have been provided for Figures 3, 4 and 6.

The following previously published datasets were used:

| Author(s) | Year | Dataset title | Dataset URL | Database and Identifier |
|---|---|---|---|---|
| Koirala S, Corfas G | 2010 | Single-cell gene expression data from Bergmann glial cells of mouse cerebellum | https://www.ncbi.nlm.nih.gov/geo/query/acc.cgi?acc=GSE18617 | NCBI Gene Expression Omnibus, GSE18617 |
| Zhang Y, Chen K, Sloan SA, Scholze AR, Caneda C, Ruderisch N, Deng S, Daneman R, Barres BA, Wu JQ | 2014 | An RNA-Seq transcriptome and splicing database of neurons, glia, and vascular cells of the cerebral cortex | https://www.ncbi.nlm.nih.gov/geo/query/acc.cgi?acc=GSE52564 | NCBI Gene Expression Omnibus, GSE52564 |

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
