## [Decision Letter]

Thank you for submitting your article "eIF2B activator prevents neurological defects caused by a chronic Integrated Stress Response" for consideration by *eLife*. Your article has been reviewed by three peer reviewers, and the evaluation has been overseen by Joseph Gleeson as the Reviewing Editor and Catherine Dulac as the Senior Editor. The following individuals involved in review of your submission have agreed to reveal their identity: Raphael Schiffmann (Reviewer #1); Yuanyi Feng (Reviewer #2).

The reviewers have discussed the reviews with one another and the Reviewing Editor has drafted this decision to help you prepare a revised submission.

Summary:

Wong et al. provide convincing evidence that the eIF2B activator (also termed stapler) 2BAct, a newly synthesized brain-permeable eIF2B activator compound, initiated in around age 6 weeks effectively prevents behavioral, pathological as well as gene expression and proteomics abnormalities in a mouse model of eIF2B-related disorder (variably termed VWM or CACH). The authors recently have shown that an eIF2B activator ISRIB can boost eIF2B activity and attenuate ISR in vitro.

This study demonstrates the efficacy 2BAct in vivo in a newly-generated VWM mouse model (R191H mice). VWM is an autosomal recessive disorder caused by mutations of eIF2B, a GEF for the translational initiator eIF2 and an inhibitor of eIF2's phosphorylation. The main pathology of VWM involves myelin loss, leading to multiple progressive neurological symptoms or death. Given that eIF2a phosphorylation is the core event for integrated stress response (ISR), which reduce protein synthesis and that IRS activation has been observed in VWM postmortem specimen and is considered as a maladaptive response due to eIF2B deficiency, restoring eIF2B function is thus expected to suppress the maladaptive ISR in VWM and ameliorate VWM symptoms. Data presented by the manuscript show that 2BAct can restore the body weight, improve motor functions, and reduce myelin loss and reactive gliosis of R191H mice. mRNA/transcriptomic analyses demonstrated that 2BAct attenuated the elevated ISR and reduced ATF4 target gene expression in R191H mice. TMT-MS data also support the reduced ISR and largely rescued brain proteome by 2BAct. These data are quite compelling and together support the efficacy of 2BAct in vivo. They demonstrate the feasibility of developing eIF2B activators to treat VWM and can also provided new insight into the pharmacological modulation of ISR signaling in other pathological conditions, such as inflammation, diabetes, cancer and neurodegenerative diseases. The data sets appear to be complete and will likely be of use and interest to the broader community.

Essential revisions:

1) It seems that at least as far as the onset of the disease is concerned it is the lower rate of protein synthesis that causes the disease rather than abnormal stress response. The authors variably refer to one or the other. Cabilly et al. Plos One 2012 suggests that rapid protein synthesis is a major problem. The lack of significant upregulation of gadd34 in the first 2.5 also supports protein synthesis defect as the problem here rather than maladaptive stress response. Wong et al. also importantly found that the process of myelination is not source for ER stress. But in subsection “A chronic ISR in the CNS of VWM mice is prevented by 2BAct” – it is not surprising that there is no ISR since maximal eIF2B GEF activity should eliminate ISR. On the other hand, in the first paragraph of the Discussion they emphasize the role of ATF4 target genes as a main culprit in causing CNS dysfunction. So is it global protein synthesis or upregulation/downregulation of ISR genes or both?

2) What will happen upon exposure to an exogenous stressor in the presence of ISRIB, which acts to avoid ISR? Will it prevent cellular rescue? The activator/stapler eliminates one aspect that is the inappropriate stress response due to the mutation but possibly also the 'appropriate' stress response associated with inhibition by eIF2-p. Will the authors perform such experiments in the near future (I am not asking for additional such experiments for this paper)? If the activation by an ISRIB is potent, why wouldn't it lead to cell death when exposed to exogenous stressors.

3) Why did the author perform almost all their experiments on the cerebellum (and spinal cord), not a major part of the brain affected by this disease (and not the cerebrum)? Is this a typo? The word 'cerebrum' does not appear anywhere in the text. What happens to the cerebral tissue?

4) This 'therapeutic approach' will only work when the effect of the mutation is mainly through de-stabilizing the decameric complex. If the mutation disturbs (is close to) the active site, it may not respond to ISRIB. Authors should mention the frequency and location of the various human mutations, to give readers a sense of the therapeutic potential.

5) To further improve the manuscript, I would like to see more quantitative measurements of gliosis, neuroinflammation, and myelination rather than the IHC data presented in Figure 2. It will also be better to include additional data to show the effect of 2BAct on WT mice and/or WT cells with respect to ISR, as this will further validate the target specificity and therapeutic safety of 2BAct.

---

## [Author Response]

Essential revisions:1) It seems that at least as far as the onset of the disease is concerned it is the lower rate of protein synthesis that causes the disease rather than abnormal stress response. The authors variably refer to one or the other. Cabilly et al. Plos One 2012 suggests that rapid protein synthesis is a major problem. The lack of significant upregulation of gadd34 in the first 2.5 also supports protein synthesis defect as the problem here rather than maladaptive stress response. Wong et al. also importantly found that the process of myelination is not source for ER stress. But in subsection “A chronic ISR in the CNS of VWM mice is prevented by 2BAct” – it is not surprising that there is no ISR since maximal eIF2B GEF activity should eliminate ISR. On the other hand, in the first paragraph of the Discussion they emphasize the role of ATF4 target genes as a main culprit in causing CNS dysfunction. So is it global protein synthesis or upregulation/downregulation of ISR genes or both?

The two outputs of the ISR, protein synthesis and induction of ATF4 targets, are tightly coupled. Reduction of ternary complex by inhibition of eIF2B activity both reduces global protein synthesis and enables the uORF-driven mechanism that promotes ATF4 ORF translation. However, the sensitivity of the two outputs differs – a small reduction in ternary complex concentration causes a small decrease in global protein synthesis, yet it is sufficient to trigger a switch-like response in ATF4 expression as well as expression of other translational targets. Under most conditions of physiological stress (including VWM), the reduction in global mRNA translation is likely small. Thus, we propose that it is the chronic ISR activation that is the major driver of progressive CNS pathology. However, a combined effect of translational induction of ISR targets such as ATF4 and ATF5 and the reduction of bulk protein synthesis rates could be driving the pathological changes.

Cabilly et al. used a mouse model of VWM (eIF2B5^R132H^) that has a very mild phenotype. They used LPS, which is a severe systemic stress, to reveal a deficiency in these mice. Under these conditions, the contribution of reduced protein synthesis to a particular phenotype is likely to be more significant. This might also be the case for patients who encounter acute stresses.

We generated the VWM mouse model used by Cabilly et al. and have included new data analyzing the effect of 2BAct in these mice (Figure 3—figure supplement 3). We measured a very small induction of the ISR in their brains, consistent with the mild behavioral phenotype and normal lifespan observed in these mice (Geva et al., 2010). As is the case for R191H, 2BAct fully normalized the expression of ISR targets in this model.

2) What will happen upon exposure to an exogenous stressor in the presence of ISRIB, which acts to avoid ISR? Will it prevent cellular rescue? The activator/stapler eliminates one aspect that is the inappropriate stress response due to the mutation but possibly also the 'appropriate' stress response associated with inhibition by eIF2-p. Will the authors perform such experiments in the near future (I am not asking for additional such experiments for this paper)? If the activation by an ISRIB is potent, why wouldn't it lead to cell death when exposed to exogenous stressors.

The reviewers raise an important consideration for all researchers working with ISR inhibitors (or small molecule eIF2B activators), as well as for the clinical development of this class of molecules.

Ourin vitro data suggest that high levels of phospho-eIF2α suppress eIF2B activity even in the presence of ISRIB (Wong et al., 2018). This is likely due to the sequestration of all available eIF2B by the high concentration of phospho-eIF2α in the assay. As eIF2 is in excess of eIF2B in cells, this is a plausible scenario in vivo. However, whether various physiological stressors produce sufficiently high level of eIF2α phosphorylation remains to be explored.

If a stress is sufficiently high as to block the effect of an eIF2B activator, we would expect the “appropriate” stress response to continue functioning. However, if the stress does not reach the threshold, eIF2B activators could exacerbate the effect, as suggested by the reviewer. The eIF2:eIF2B ratio, as well as the amount of eIF2α phosphorylated, is likely to be cell-type dependent. Thus, the threshold at which different cells/tissues will experience a potential loss of eIF2B activator’s effects will be variable. In the future, we plan to conduct further experiments to examine this question.

3) Why did the author perform almost all their experiments on the cerebellum (and spinal cord), not a major part of the brain affected by this disease (and not the cerebrum)? Is this a typo? The word 'cerebrum' does not appear anywhere in the text. What happens to the cerebral tissue?

We performed targeted gene expression analysis on all regions of the brain (cerebellum in Figure 3A; forebrain, midbrain and hindbrain of cerebrum in Figure 3—figure supplement 1A) and found little difference between the regions with respect to ISR induction. Based on this, we used the cerebellum for further analysis for four reasons:

1) Cerebellar ataxia is a classical phenotype of the disease (van der Knaap et al., 2006; Bugiani et al., 2010)

2) Bergmann glia are reported to be involved in VWM pathology (Dooves et al., 2017).

3) Ease of isolation of the tissue

4) Reduced chance of confounding results as it does not involve a “mixture” of brain regions

We analyzed the spinal cord as it has been reported to show significant pathology in both mutant mice as well as VWM patients (Meoded et al., *Neuropediatrics* 2011;Leferink et al., 2018). Our spinal cord data are consistent with existing evidence, as we observed the greatest degree of ISR induction and histological pathology in this tissue.

Finally, we have added new single-cell RNA-seq data from two brain regions, forebrain and cerebellum (Figure 5 and Figure 5—figure supplements 1-3). Notably, our analysis implicates astrocyte populations (and specifically Bergmann glia in the cerebellum) as the primary source of ISR activation in both tissues. To our knowledge, this represents the first unbiased identification of astrocytes as the likely causal cell population in this VWM mouse model, and corroborates IHC findings by others. We also observe lesser but significant ISR induction in other non-neuronal cell types, suggesting that these cells may also contribute to disease progression. Notably, the basal expression level of ISR targets is higher in the astrocyte population than in other cell types, which may explain their heightened sensitivity to the eIF2B mutation. These findings will guide further studies with human post-mortem tissues to delve into the causal cell populations in the human disease.

4) This 'therapeutic approach' will only work when the effect of the mutation is mainly through de-stabilizing the decameric complex. If the mutation disturbs (is close to) the active site, it may not respond to ISRIB. Authors should mention the frequency and location of the various human mutations, to give readers a sense of the therapeutic potential.

We previously showed that eIF2B activators can boost the GEF activity of five different VWM mutations in various subunits of the eIF2B complex, including one (eIF2B5^R136H^) that did not have apparent in vitro defects in decamer stability or enzymatic activity (Wong et al., 2018). As discussed in point 1 above, 2BAct also rescued the mild ISR of the eIF2B5^R132H^ mutant in vivo. Thus, eIF2B activators have worked on all mutations tested to date, including one whose defect is not captured by our in vitro assays.

It is possible to introduce non-naturally occurring mutations that block ISRIB binding to eIF2B (Sekine et al., 2015; Wong et al., 2018; Tsai et al., 2018). However, to date, no VWM mutations have been identified in the ISRIB/2BAct-binding region of eIF2B. We have updated the last paragraph of our Discussion section to encompass these points.

5) To further improve the manuscript, I would like to see more quantitative measurements of gliosis, neuroinflammation, and myelination rather than the IHC data presented in Figure 2. It will also be better to include additional data to show the effect of 2BAct on WT mice and/or WT cells with respect to ISR, as this will further validate the target specificity and therapeutic safety of 2BAct.

We measured myelination in two independent ways (Luxol Fast Blue and MBP staining) in both the cerebellum and spinal cord (Figure 2 and Figure 2—figure supplement 1; spinal cord MBP data newly added). We also used two different markers each to measure gliosis (GFAP and Olig2) and inflammation (ATF3 and Iba-1). We note that the magnitudes of our observed changes are large, and given the large number of samples analyzed (6 males and 6 females per condition), highly significant. Thus, we believe that our data represent accurate quantitative measurements. Furthermore, our RNA-seq data (Figure 3 and Supplementary File 1C) revealed decreased expression of myelination- and glial-related genes, which was rescued by 2BAct. Collectively, we believe the data are all internally consistent and adequately support our conclusions.

With respect to the target specificity of eIF2B activators, we have included new data (Figure 4—figure supplement 2) comparing gene expression in placebo- and 2BAct-treated WT mice. The heatmap of RNA-seq data shows that 2BAct-treated mice have similar transcriptomes to and do not cluster away from placebo-treated mice. The volcano plot reiterates this point by plotting the differential expression of 2BAct-treated vs. placebo-treated samples. Thus, eIF2B activators do not elicit gene expression changes under normal conditions. We re-emphasize that 2BAct itself is not a suitable therapeutic for humans due to CV liabilities. Other eIF2B activators that may be generated in the future will require in vivo testing for on- and off-target effects on a case-by-case basis.